# Prior Learning in Introspective VAEs

**Ioannis Athanasiadis** *ioannis.athanasiadis@liu.se*
*Department of Electrical Engineering*
*Linköping University*

**Fredrik Lindsten** *fredrik.lindsten@liu.se*
*Department of Computer and Information Science*
*Linköping University*

**Michael Felsberg** *michael.felsberg@liu.se*
*Department of Electrical Engineering*
*Linköping University*

**Reviewed on OpenReview:** *https://openreview.net/forum?id=u4YDVFodYX*

## Abstract

Variational Autoencoders (VAEs) are a popular framework for unsupervised learning and data generation. A plethora of methods have been proposed focusing on improving VAEs, with the incorporation of adversarial objectives and the integration of prior learning mechanisms being prominent directions. When it comes to the former, an indicative instance is the recently introduced family of Introspective VAEs aiming at ensuring that a low likelihood is assigned to unrealistic samples. In this study, we focus on the Soft-IntroVAE (S-IntroVAE), one of only two members of the Introspective VAE family, the other being the original IntroVAE. We select S-IntroVAE for its state-of-the-art status and its training stability. In particular, we investigate the implication of incorporating a multimodal and trainable prior into this S-IntroVAE. Namely, we formulate the prior as a third player and show that when trained in cooperation with the decoder constitutes an effective way for prior learning, which shares the Nash Equilibrium with the vanilla S-IntroVAE. Furthermore, based on a modified formulation of the optimal ELBO in S-IntroVAE, we develop theoretically motivated regularizations, namely (i) adaptive variance clipping to stabilize training when learning the prior and (ii) responsibility regularization to discourage the formation of inactive prior modes. Finally, we perform a series of targeted experiments on a 2D density estimation benchmark and in an image generation setting comprised of the (F)-MNIST and CIFAR-10 datasets demonstrating the effect of prior learning in S-IntroVAE in generation and representation learning.

## 1 Introduction

Variational Autoencoders (VAEs) (Rezende et al., 2014; Kingma & Welling, 2013) constitute a popular generative framework where variational inference is utilized to learn low-dimensional embeddings by modeling the density of the high-dimensional data. VAEs enjoy a plethora of applications, ranging from anomaly detection (Chauhan et al., 2022) to representation disentanglement (Higgins et al., 2017) and high-resolution image generation (Razavi et al., 2019).

From a representation learning perspective, VAEs produce structured latent spaces due to the regularization imposed by fitting a prior distribution. This contrasts with unregularized autoencoders, which often lack such structure (Shrivastava et al., 2024; Oring, 2021; Leeb et al., 2020). These structured representations are particularly valuable in domains like scientific discovery, where understanding the underlying data structure is critical (Wang et al., 2023).

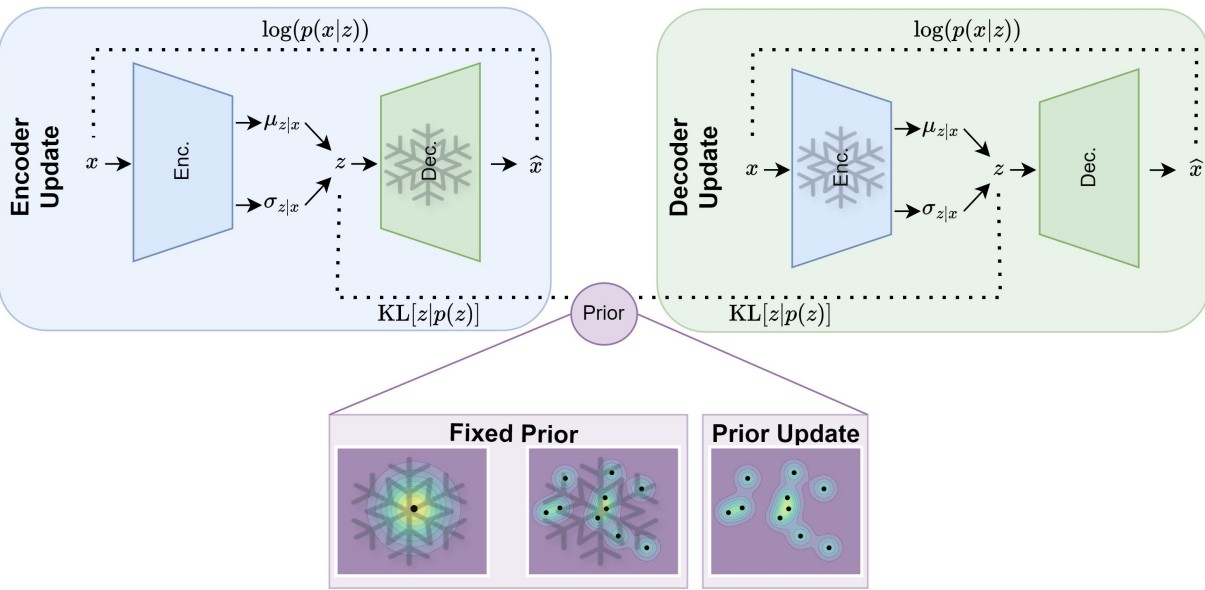

Figure 1: Prior player realizations within Introspective VAEs. The prior component can be regarded as a third player that can actively participate in the adversarial game along with the encoder and the decoder. The overlayed snowflake indicates that the component is not updated.

Despite VAEs falling short of other popular generative paradigms, such as the Generative Adversarial Networks (GANs) (Goodfellow et al., 2020) and diffusion models (Ho et al., 2020) in terms of generation quality, they are distinctive in the sense of simultaneously providing the amortized inference and generation modeling (Gatopoulos & Tomczak, 2021). Building upon that, combining VAEs with these frameworks has been a popular research direction aiming at retaining its merits while mitigating its limitations (Makhzani et al., 2016). With diffusion models emerging as the state-of-the-art framework for image synthesis (Yang et al., 2022), there are recent works on VAE/diffusion hybrids (Preechakul et al., 2022; Rey et al., 2019). Additionally, the VAE/GAN hybrid literature is also an established sub-field targeted at improving the poor training stability of GANs (Mescheder et al., 2018) and generation diversity (Huang et al., 2018) while addressing the blurry generation of VAEs.

Another independent direction to improving VAEs is learning the prior as opposed to fitting into a fixed one, commonly the standard Gaussian. Trainable priors allow for identifying structure within the data, which is of high interest in the unsupervised and semi-supervised learning setups. Moreover, sufficiently expressive priors are needed for generating realistic data from complex distributions (Lavda et al., 2019; Dilokthanakul et al., 2016). Additionally, utilizing the structured capture in the latent space (Lavda et al., 2019) can benefit the generation performance as well as provide control over the semantics of the generated samples even in the absence of labels.

Motivated by the prospect of combining the strength of two distinct and conceptually different directions for enhancing VAEs, we consider the problem of incorporating prior learning in the S-IntroVAE (Daniel & Tamar, 2021) framework. Our intuition is that the appealing features of reducing over-regularization and holes as enabled by prior learning are not sufficient for realistic sample generation. On the other hand, although adversarially trained VAEs possess higher quality generation capabilities, they are still subject to the problem associated with assuming an over-simplistic prior.

Based on these, we formulate the prior as an additional player in S-IntroVAE which participates in the adversarial training. More specifically, we extend the original analysis provided by Daniel & Tamar (2021) and conclude that the prior–decoder cooperation scheme is a viable option for learning the prior while remaining faithful to the Nash Equilibrium (NE) of the vanilla S-IntroVAE. Our work is partly related to the CS-IntroVAE (Yu et al., 2023), where a fixed three-component Mixture of Gaussian (MoG) prior was

integrated into S-IntroVAE by replacing the Kullback–Leibler (KL) with the Cauchy–Schwarz divergence to allow for closed-form divergence computation. Notably, in our work, we follow the original variational analysis provided in Daniel & Tamar (2021), using the KL divergence and its theoretical properties, thereby investigating the effect of using a multimodal prior, including its trainable form, in isolation. Formally our contributions are:

- extending the original S-IntroVAE under the prior–decoder cooperation scheme.
- two theoretically motivated regularizations (i) adaptive variance clipping and (ii) responsibilities entropy, which enable robust prior learning.
- the experiments on a synthetic 2D density estimation and an image generation task demonstrating the effect of prior learning in S-IntroVAE in generation and representation learning.

## 2 Related Work

**VAEs:** In VAEs (Rezende et al., 2014; Kingma & Welling, 2013) an autoencoder-based structure is utilized, along with variational inference, to maximize a lower bound on the marginal log-likelihood of the data (the evidence lower bound, ELBO). More specifically, this resorts to simultaneously minimizing the sum of the empirical reconstruction error and the Kullback–Leibler (KL) divergence between the extracted latent representations and an assumed prior (typically the standard Gaussian distribution). A tighter ELBO was proposed by Burda et al. (2015), based on an importance weighting scheme, providing more flexibility during training by being more forgiving of inaccurate posterior estimates. Hierarchical variations of VAEs (Vahdat & Kautz, 2020; Sønderby et al., 2016) rely on multiple stochastic layers where each of them is conditioned on the previous one, resulting in more efficient representation learning (Child, 2020; Zhao et al., 2017).

**Prior Assumption in VAEs:** Several studies suggest that assuming an over-simplistic prior can over-regularize the VAEs hindering their performance (Lin & Clark, 2020; Tomczak & Welling, 2018; Hoffman & Johnson, 2016). Goyal et al. (2017) argue that assuming a standard Gaussian prior can omit meaningful semantic information in the latent representation. Moreover an over-simplistic prior introduces holes in the prior negatively affecting the generation capabilities of VAEs (Aneja et al., 2021; Rezende & Viola, 2018). Towards addressing this shortcoming, Tomczak & Welling (2018) proposed the VampPrior where trainable pseudo-inputs are fed into the encoder providing the parameters of a MoG distribution to replace the standard one. Connor et al. (2021) adopt a manifold-learning approach to define an MoG prior, which is better crafted for the latent space of the data. Kalatzis et al. (2020) assume a Riemannian latent space where the prior is inferred from the data, replacing the standard Gaussian with a Brownian motion prior.

**Adversarial Objectives in VAEs:** In Adversarial Autoencoders (AEEs) (Makhzani et al., 2016) the latent space is regularized into following the assumed prior through a min-max game between the encoder and a discriminator module. The VAE/GAN hybrid was proposed by Larsen et al. (2016) where the similarity distance, for measuring the reconstruction error, is implicitly learned through an adversarial game in which the decoder network serves as both a VAE decoder and the generator of a GAN. In the seminal IntroVAE (Huang et al., 2018), the VAEs are framed as an adversarial game between the encoder and the decoder by considering the KL divergence as an energy function. The S-IntroVAE improves the training stability of IntroVAE, while also providing the theoretical analysis suggesting that Introspective VAEs constitute a variational instance of GANs. In CS-IntroVAE (Yu et al., 2023) the KL was replaced by Cauchy–Schwarz divergence while using a fixed three-component MoG in S-IntroVAE leading to improved generation performance.

## 3 Background

Our work builds upon the framework proposed by Daniel & Tamar (2021). To avoid confusion we adopt, whenever possible, identical notations as presented in their work. Let $x \sim p_{\text{data}}(x)$ be a data sample and $z$ its latent representation. A VAE aims at learning a parametric model $p_{d_\theta}(x, z) = p_{d_\theta}(x|z)p_z(z)$ such that the marginal log-likelihood of the data is maximized. Due to the intractability of that likelihood (Kingma & Welling, 2013), we resort to maximizing the ELBO. Assuming a prior $p_z$ on the latent space, an encoder $q_\phi$

providing the approximating posterior and a decoder $p_{d_\theta}$, parametrized by $\phi$ and $\theta$ respectively, we evaluate the ELBO, denoted as $W$, at point $x$ as:

$$W(x; q_\phi, p_{d_\theta}) = -\text{KL}[q_\phi(z|x)||p_z(z)] + \mathbb{E}_{z \sim q_\phi(z|x)}[\log p_{d_\theta}(x|z)] \leq \log p(x) \tag{1}$$

with $\text{KL}[\cdot||\cdot]$ denoting the KL divergence. In practice, the encoder and the decoder are typically realized through neural networks with parameters $\phi$ and $\theta$, respectively. The $\beta$-VAE reformulates the ELBO by weighting the relative contribution of the KL term using the $\beta$ hyperparameter, that is:

$$W(x; q_\phi, p_{d_\theta}, \beta) = -\beta \cdot \text{KL}[q_\phi(z|x)||p_z(z)] + \mathbb{E}_{z \sim q_\phi(z|x)}[\log p_{d_\theta}(x|z)]. \tag{2}$$

Note that, from an optimization perspective, the $\beta$-VAE ELBO formulation is equivalent to using independent weighting hyperparameters for each of its constituting terms, such that $W(x; q_\phi, p_{d_\theta}, \beta_{\text{rec}}, \beta_{\text{KL}}) = -\beta_{\text{KL}} \cdot \text{KL}[q_\phi(z|x)||p(z)] + \beta_{\text{rec}} \cdot \mathbb{E}_{z \sim q_\phi(z|x)}[\log p_{d_\theta}(x|z)]$. This ELBO formulation is convenient for tuning the S-IntroVAE and therefore is the adopted ELBO formulation. Additionally, when clear from the context, we omit expressing the ELBO with respect to the $\beta_{\text{KL}}$ and $\beta_{\text{rec}}$ to enhance clarity.

## 3.1 Learning the optimal prior

In light of the previously discussed implication of imposing a simple prior in VAE, the question arises: what is the optimal prior $p_z(z)$? In this aspect, an insightful reformulation of the empirical ELBO is provided by Tomczak (2022):

$$\begin{aligned}
\mathbb{E}_{x \sim p_{\text{data}}(x)}[W(x; q_\phi, p_{d_\theta}, \beta_{\text{rec}}, \beta_{\text{KL}})] = \beta_{\text{rec}} \cdot \mathbb{E}_{x \sim p_{\text{data}}(x)}[\mathbb{E}_{z \sim q_\phi(z|x)}[\log p_{d_\theta}(x|z)] \\
+ \beta_{\text{KL}} \cdot (\mathbb{H}[q_\phi(z|x)]] - \mathbb{CE}[q_\phi(z)||p_z(z)]),
\end{aligned} \tag{3}$$

with $\mathbb{H}[\cdot]$ and $\mathbb{CE}[\cdot||\cdot]$ denoting the Shannon and the cross-entropies, respectively, and $q_\phi(z) = \mathbb{E}_{x \sim p_{\text{data}}(x)}[q_\phi(z|x)]$ is the aggregated posterior. The formulation above suggests that the optimal prior can be found as the maximizer of the ELBO, namely $p_z(z) = q_\phi(z)$, as this is when the negative cross entropy term is maximized (Gibbs' inequality). Towards this, utilizing a trainable MoG prior emerges as a relevant alternative to the standard Gaussian. A prior–encoder pairing was realized by Tomczak & Welling (2018), termed as VampPrior, leading to better separation in latent space. Formally, the MoG and Vamp $M$-modal priors, denoted by $p_\lambda(z)$ and $p^q(z)$ respectively, are parametrized as:

$$p_\lambda(z) = \sum_{i=1}^{M} w_i \cdot \mathcal{N}(z|\mu_i, \sigma_i^2 I) \quad \text{and} \quad p^q(z) = \sum_{i=1}^{M} w_i \cdot q_\phi(z|x_i), \tag{4}$$

with $\sum_{i=1}^{M} w_i = 1$ and $w_i$ the contribution of each component, $\mu_i$ and $\sigma_i$ the means and variances of the MoG prior and $x_i$ pseudo-inputs for the VampPrior.

## 3.2 S-IntroVAE

correIn typical VAEs, the encoder and the decoder are updated simultaneously in a single backpropagation stage. Motivated by the observation that assigning a high likelihood for the real data does not necessarily imply assigning a low likelihood for the unlikely ones, the Introspective VAEs family (Daniel & Tamar, 2021; Huang et al., 2018) formulates an adversarial game between the encoder and the decoder. In S-IntroVAE (Daniel & Tamar, 2021), the ELBO is regarded as an energy function, and on that basis, the encoder is induced to assign high energy to real and low energy to generated data. On the contrary, the decoder aims at generating data (i.e., reconstructed and generated samples) that resemble those of the real data distribution to fool the encoder. The above setup constitutes an adversarial game between the encoder and the decoder similar to the GAN (Goodfellow et al., 2020) paradigm.

For notational brevity in the derivations below we drop the dependence on the parameters $\theta$ and $\phi$ and simply write $d$ for the decoder and $q$ for the encoder, while we henceforth refer to $\mathbb{E}_{x \sim p(x)}[\cdot]$ simply as $\mathbb{E}_p[\cdot]$ when clear from the context. Formally, given the empirical $p_{\text{data}}(x)$ and $p_d(x) = \mathbb{E}_{p_z(z)}[p_d(x|z)]$ the generated data distribution, the encoder $q$ and decoder $d$ are alternately updated towards maximizing their respective objectives $L_q(q, d)$ and $L_d(q, d)$ defined as:

$$
\begin{aligned}
L_q(q, d) &= \mathbb{E}_{p_{\text{data}}}\left[W(x; q, d)\right] - \mathbb{E}_{p_d}\left[\frac{1}{\alpha} \cdot \exp(\alpha W(x; q, d))\right], \\
L_d(q, d) &= \mathbb{E}_{p_{\text{data}}}\left[W(x; q, d)\right] + \gamma \cdot \mathbb{E}_{p_d}\left[W(x; q, d)\right],
\end{aligned}
\tag{5}
$$

where $\alpha \geq 1$ and $\gamma \geq 0$ are hyperparameters. Daniel & Tamar (2021) show that there is a NE for this two-player game. Specifically, define $d^*$ as:

$$
d^* \in \arg\min_d \left\{ \text{KL}[p_{\text{data}}(x) || p_d(x)] + \gamma \cdot \mathbb{H}[p_d(x)] \right\}.
\tag{6}
$$

**Assumption 1** (Modified - (Daniel & Tamar, 2021))**.** *For all $x$ such that $p_{data}(x) \geq 0$ we have that $[p_{d^*}(x)]^{\alpha+1} \leq p_{data}(x)$.*

**Theorem 1** ((Daniel & Tamar, 2021))**.** *Under the Assumption 1, the pair of optimal $q^* = p_{d^*}(z|x)$ and $d^*$ as defined in* (6) *constitutes a NE of the game* (5)*.*

**Remark 1.** *The Assumption 1 is a modified version of the one used by Daniel & Tamar (2021) and essentially suggests that $p_{d^*}(x)$ has to be sufficiently enclosed by the true data distribution. This modification corrects a seemingly minor oversight that, however, has important implications for the interpretation of the theorem. In particular, the proof of Theorem 1 requires expressing the condition under which the optimal $q^*$, which maximizes $L_q$, satisfies $KL[q^*(z|x) || p_d(z|x)] = 0$. Importantly, if a sample $x$ lies outside of the support of $p_{data}(x)$ but within the support of $p_d(x)$, then the optimal $q^*$ satisfies $KL[q^*(z|x) || p_d(z|x)] = \infty$. The original assumption in (Daniel & Tamar, 2021) does not separate this case, whereas the modified assumption properly accounts for it. See B.1 for a detailed explanation of this matter.*

We refer the readers to the original work of Daniel & Tamar (2021) for the proof that for every $p_{\text{data}}$ there always exists $\gamma \geq 0$ such that Assumption 1 holds for $p_{d^*}$. Theorem 1 suggests that, at convergence, the S-IntroVAE formulation leads to optimal inference capabilities (i.e., the approximated posterior equals the true one) while the generated data distribution converges to an entropy-regularized version of the true data distribution.

## 4 Prior Learning in S-IntroVAE

### 4.1 Theoretical analysis

In this section, we extend S-IntroVAE by introducing a third player dedicated to modeling the prior. Our formulation draws inspiration from DeLiGAN (Gurumurthy et al., 2017) where the noise in GANs was parametrized by a learnable MoG. In contrast to DeLiGAN, in our setting the prior (which is similar to the noise in GANs) has a dual role as (i) the source of the generated data distribution and (ii) the target based on which the adversarial training is performed. We theoretically analyze the implication of training the prior within the S-IntroVAE and conclude that learning it in cooperation with the decoder constitutes a viable option for prior learning.

In our three-player setup the encoder $q$, the decoder $d$, and the prior $\lambda$ are all flexible. We denote the generated data distribution as $p_d^\lambda(x) = \mathbb{E}_{p_\lambda(z)}[p_d(x \mid z)]$ to highlight its dependence on both the decoder $d$ and the prior $\lambda$ players. In that case, the adversarial game of (5) becomes:

$$
\begin{aligned}
L_q(\lambda, q, d) &= \mathbb{E}_{p_{\text{data}}}\left[W(x; \lambda, q, d)\right] - \mathbb{E}_{p_d^\lambda}\left[\frac{1}{\alpha} \cdot \exp(\alpha \cdot W(x; \lambda, q, d))\right], \\
L_d(\lambda, q, d) &= \mathbb{E}_{p_{\text{data}}}\left[W(x; \lambda, q, d)\right] + \gamma \cdot \mathbb{E}_{p_d^\lambda}\left[W(x; \lambda, q, d)\right].
\end{aligned}
\tag{7}
$$

The encoder is trained to maximize the $L_q$ whereas the prior and the decoder maximize the $L_d$ objective (i.e., prior–decoder cooperation). Below we show that prior–decoder cooperation is a viable option for prior learning which retains NE from the original S-IntroVAE formulation.

We modify (6) to support our learnable prior setup. Let $\Lambda$ denote the set of possible parameterizations of the prior and $\lambda \in \Lambda$, we define:

$$(\lambda^*, d^*) \in \underset{\lambda, d}{\arg\min} \left\{ \mathrm{KL}[p_{\text{data}}(x) || p_d^\lambda(x)] + \gamma \cdot \mathbb{H}[p_d^\lambda(x)] \right\}. \tag{8}$$

Let us also extend Assumption 1 to account for the prior being learnable.

**Assumption 2.** *For all $x$ such that $p_{data}(x) \geq 0$ we have that $[p_{d^*}^{\lambda^*}(x)]^{\alpha+1} \leq p_{data}(x)$.*

**Corollary 1.** *Under the Assumption 2, when training the prior player $\lambda$ in cooperation with the decoder player $d$ then the triplet $q^* = p_{d^*}^{\lambda^*}(z|x)$, $\lambda^*$ and $d^*$ as defined in (8) constitutes a NE of the game (7).*

*Sketch of proof.* The proof of Corollary 1 follows from Theorem 1, under the modified Assumption 1 (see Remark 1). Analogous to Theorem 1 ((Daniel & Tamar, 2021)), proving Corollary 1 entails first showing that the optimal encoder converges to the true posterior under the Assumption 2. Then, given that the encoder has converged, the prior and the decoder as defined in (8) maximize the $L_d$ objective as defined in (7). This concludes the proof of Corollary 1. For the complete proof, we refer readers to B.2.2. □

Our three-player formulation is similar in nature to S-IntroVAE with the encoder converging to the true posterior while the generated data distribution converges to an entropy-regularized version of the real data distribution. The key difference, however, lies in the fact that our formulation allows for a trainable prior, unlocking the merits of prior learning such as mitigating the prior hole problem, unsupervised clustering (Dilokthanakul et al., 2016), explainability (Klushyn et al., 2019), and more controllable generation (Lavda et al., 2019). More specifically, for fixed encoder $q$ and decoder $d$, given a batch of real and generated data respectively, the prior update seeks (i) to support a linear combination (controlled by the $\gamma$ hyperparameter) of the empirical real and fake aggregated posterior and (ii) be idempotent under the projection by $d$.

### 4.1.1 Optimal ELBO in the assumption-free setting

Corollary 1 requires Assumption 2 to hold, however, in practice this might not be the case, especially early in training. For instance, having a $p_d^\lambda(x)$ generating (i) out-of-distribution data or (ii) realistic samples at a disproportionately higher rate compared to the real distribution, are two obvious cases where such an assumption is violated. Analyzing the behavior of the encoder in these cases provides an intuitive connection to regularly trained VAEs and motivates some of our implementation choices. Let $\mathbb{X} = \{x | x \in p_{\text{data}}(x) > 0 \cup p_d^\lambda(x) > 0\}$ (i.e., the set of all possible samples in the union of real and generated data supports), we define the ELBO $W(x; \lambda, q^*, d)$ as:

$$W(x; \lambda, q^*, d) = \begin{cases} -\infty, \ x \in \{x \in \mathbb{X} \mid p_{\text{data}}(x) = 0\} \\ \\ \frac{1}{\alpha} \cdot \log \frac{p_{\text{data}}(x)}{p_d^\lambda(x)}, \ x \in \{x \in \mathbb{X} \mid p_{\text{data}}(x) > 0 \cap [p_d^\lambda(x)]^{\alpha+1} > p_{\text{data}}(x)\} \\ \\ \log p_d^\lambda(x), \ x \in \{x \in \mathbb{X} \mid p_{\text{data}}(x) > 0 \cap [p_d^\lambda(x)]^{\alpha+1} \leq p_{\text{data}}(x)\}\} \end{cases} \tag{9}$$

**Proposition 1.** *Given a fixed generated data distribution $p_d^\lambda(x)$ the $q^*$ maximizing $L_q(\lambda, d, q)$ in Eq. 7 is such that the ELBO $W(x; \lambda, q^*, d)$ satisfies Eq. 9.*

The proposition (for the proof see B.3) above suggests that under the Assumption 2 the encoder in S-IntroVAE behaves similar to the one in regular VAEs. Alternatively, as a consequence of the repelling objective acting on the generated data, the encoder in S-IntroVAE diverges from its VAE-optimal state. This

divergence depends on the sample-wise mismatch between $p_d^\lambda(x)$ and $p_{\text{data}}(x)$. Interestingly, it also appears that the optimal ELBO with respect to the encoder is a continuous function of the $p_{\text{data}}(x)$ measure.

### 4.1.2 Practical implications in the assumption-free setting

Let us now investigate how the theoretical claims suggested by Proposition 1 are realized in practice. For this purpose, the image generation setting was deemed an appropriate testbed due to being easy to interpret while at the same time sufficiently complex allowing us to draw generalizable conclusions. Note that proposition only concerns the optimal encoder given fixed real and generated distributions. Based on that, we employ a well-trained S-IntroVAE and overfit the encoder network while keeping the prior and decoder fixed. In this regard, having the prior and the decoder fixed translates to having a fixed generated data distribution.

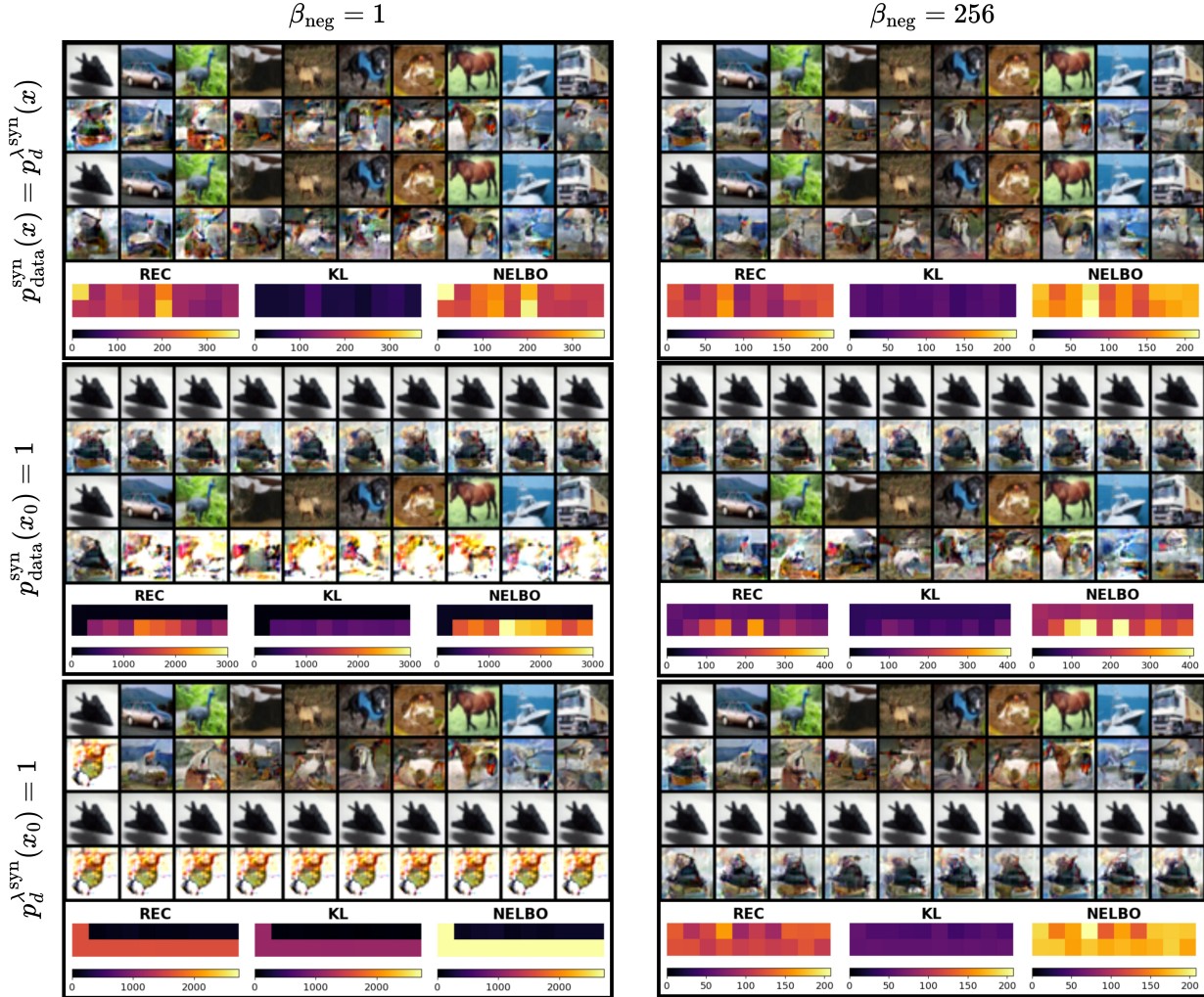

Figure 2: Overfitting the encoder given a fixed generated data distribution across three different configurations (rows). The experiment was conducted both under the theoretically faithful hyperparameter setting ($\beta_{\text{neg}} = 1$ - left) and the one used in practice ($\beta_{\text{neg}} = 256$ - right). The first line in the REC, KL and NELBO plots refers to the real data distribution whereas the second one refers to the generated data distribution. The image data figures, in each configuration, correspond to the real data distribution, the reconstructed real data distribution, the generated data distribution and the reconstructed generated data distribution from top to bottom. The figures above were generated by utilizing a trained S-IntroVAE trained under a fixed 10-modal MoG prior.

As outlined by the proposition, the encoder treats each sample $x$ (i.e., image in this context) differently depending on the likelihood ratio between $p_{\text{data}}(x)$ and $p_d^\lambda(x)$. Unfortunately, to this end, we can not make use of the Proposition 1 as we do not have access to the analytical densities of either of these distributions. To overcome the aforementioned challenge we use a subset of the real data distribution to construct synthetic real and generated data distributions, denoted as $p_{\text{data}}^{\text{syn}}(x)$ and $p_d^{\lambda^{\text{syn}}}(x)$ respectively. Additionally, it was also necessary, to use a batch size of 1 to avoid leaking information between samples inside and outside of the support of real data distribution due to the batch normalization layers. Based on these synthetic distributions, we can use them as proxies for testing the proposition. Specifically, we experiment with three distinct configurations with different properties: (i) $p_{\text{data}}^{\text{syn}}(x) = p_d^{\lambda^{\text{syn}}}(x)$ where both distributions consist of multiple different samples (ii) $p_{\text{data}}^{\text{syn}}(x)$ consisting of a single sample $x_0$, whereas $p_d^{\lambda^{\text{syn}}}(x)$ consists of multiple samples, including the $x_0$ of the $p_{\text{data}}^{\text{syn}}(x)$ and (iii) the reversed (ii) where $p_{\text{data}}^{\text{syn}}(x)$ and $p_d^{\lambda^{\text{syn}}}(x)$ distributions are swapped. We used 10 samples, one for each class, to construct the synthetic distributions.

In its theoretical faithful realization the results for (i), (ii) and (iii) are displayed on the left side of Figure 2 under $\beta_{\text{neg}} = 1$. These closely align with what has been suggested by the proposition where when the likelihood of generating a sample is sufficiently enclosed by the likelihood of observing that sample in the real data distribution then the encoder pushes the ELBO towards VAE-optimal levels. On the other hand, in cases where there is a significant likelihood mismatch the encoder can afford to either push the ELBO to its optimal level or diverge from that depending on whether the mismatch appears with respect to the real or the fake data distribution. For instance, when looking at configuration (ii) ($2^{\text{nd}}$ row) the encoder minimizes the NELBO (negative ELBO) for the image that is 10 times more likely under the real distribution compared to the fake distribution, whereas the NELBO increases for the samples outside the support of the real data distribution.

In practice, when computing the loss corresponding to maximizing $L_q$ objective, the real and fake ELBOs use different weights for the reconstruction and the KL losses, in particular for CIFAR-10 the $\beta_{\text{neg}}$, corresponding to the $\beta_{\text{KL}}$ for the fake ELBO, was set to 256 while the remaining $\beta's$ were set to 1. Using $\beta_{\text{neg}} = 256$ essentially prompts the encoder to focus more on the KL compared to the reconstruction loss when repelling the fake data. However, even in this case, where the hyperparameter configuration diverges from the one theoretically accounted for, we observe that similar patterns emerge.

## 4.2 Implementation

In this section, we outline the implementation choices as well as the motivation behind them enabling prior learning in S-IntroVAE in a prior–decoder cooperation manner. Pseudo-code for the prior learning in S-IntroVAE is provided in Algorithm 1.

### 4.2.1 Prior as source and target

In the prior–decoder cooperation setting the prior player $\lambda$ maximizes $L_d(\lambda, q, d)$. In practice, given a real $x_{\text{real}}$ and $z_s \sim p_\lambda(z)$, the prior minimizes the loss $L_P(x_{\text{real}}, z_s)$ given by:

$$\begin{aligned} L_P(x_{\text{real}}, z_s) =& \beta_{\text{rec}} \cdot L_{\text{rec}}(x_{\text{real}}) + \beta_{\text{KL}} \cdot L_{\text{KL}}(x_{\text{real}}) + \gamma \cdot \big( \gamma_r \cdot \beta_{\text{rec}} \cdot L_{\text{rec}}(\mathbf{sg}(D(z_s))) \\ &+ \beta_{\text{KL}} \cdot L_{\text{KL}}(D(z_s)) \big), \end{aligned} \tag{10}$$

where $D(z_s)$ is the fake sample generated from decoding the latent $z_s$, while $L_{\text{rec}}$ and $L_{\text{KL}}$ the reconstruction and the KL losses respectively. We remained consistent with the S-IntroVAE, where the reconstruction of fake data was scaled by $\gamma_r = 10^{-8}$, and the stop-gradient ($\mathbf{sg}$) operator was applied when generating a fake sample before computing its reconstruction loss. Additionally, we observe that the reconstruction loss for the real sample is not affected by the prior. In light of these, the prior player is trained both as a target for the real and fake posterior and as a source of fake samples. Based on that, a subtle issue arises when minimizing the $L_{\text{KL}}(D(z_s))$ term, since the prior can minimize it by either becoming a good source for generating realistic data or a good target that supports the posterior of generated data of low quality. The latter case is particularly problematic during the early stages of training, when the generated data lie outside the support of the real data, causing the encoder to assign a suboptimal posterior, as described in

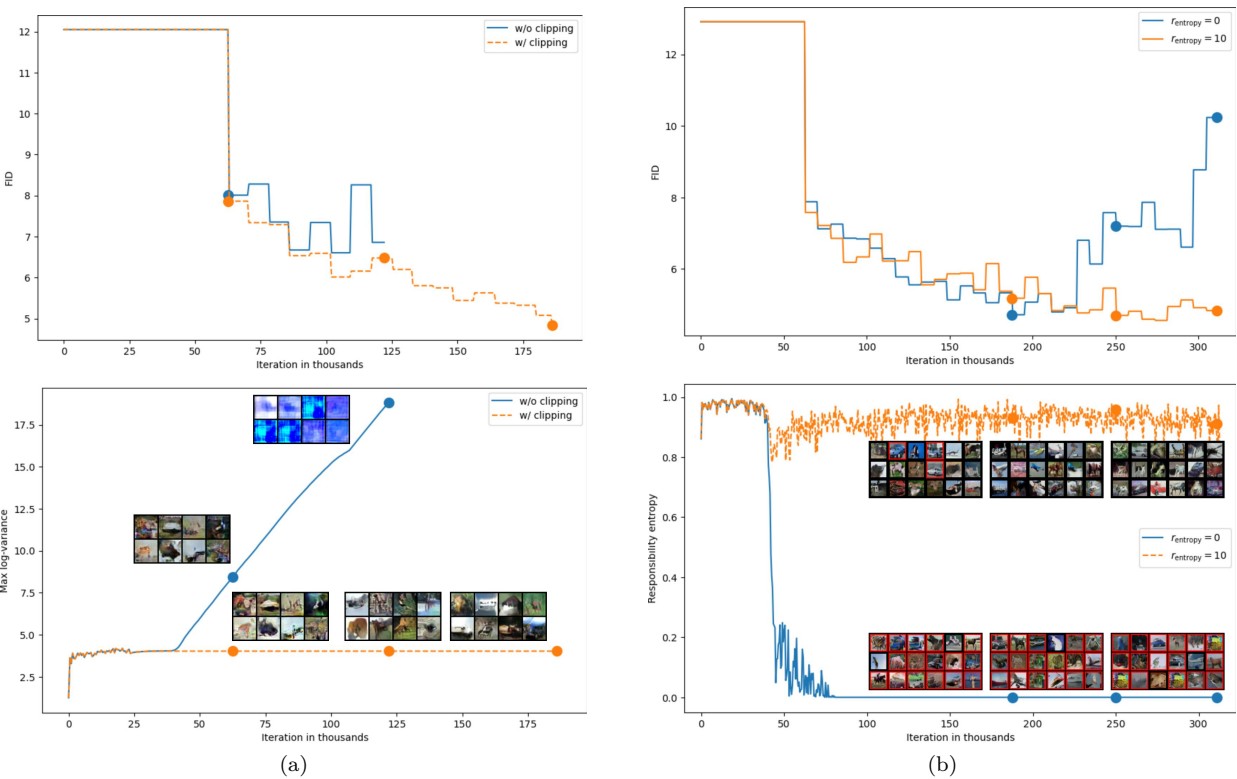

Figure 3: Regularization for robust prior learning in S-IntroVAE. (a) Clipping the prior log-variance is crucial for maintaining the stability of S-IntroVAE under the prior–decoder cooperation. Two models were trained on CIFAR-10 for 120 epochs or until crashing. Without clipping, the log-variance tends to explode, leading to an increase in the FID metric and ultimately causing the model to generate indiscernible patterns before crashing. A unimodal trainable prior was used for both models differing only by the log-variance clipping. (b) Emerging mode collapse due to unconstrained generation. We trained two models on CIFAR-10 for 200 epochs under a 10-modal MoG (log-variance clipped), differing only in the amount of entropy regularization. The red border indicates samples generated by inactive modes (i.e., average responsibility smaller than $10^{-2}$). Note that not regularizing the responsibility entropy quickly degenerates into a unimodal prior setting where a single mode is responsible for supporting the aggregated posterior. The unimodal collapse eventually leads to mode collapse and an increase in FID due to the unconstrained generation originating from the inactive modes. On the contrary, regularizing the responsibility entropy maintains more uniform responsibility allocation among the modes and addresses the mode collapse issue.

Proposition 1. To address this, we follow Shocher et al. (2023) and apply the `sg` operator to the prior as the target while allowing gradient flow for the prior as the source when computing $L_{\text{KL}}$ for the fake samples. We henceforth refer to this modified $L_{\text{KL}}$ as $L_{\text{KL}}^{\text{sg}}$ which replaces the original when computing the KL loss of the $D(z_s)$ in (10).

### 4.2.2 Adaptive variance soft-clipping

Although theoretically sound, the prior–decoder cooperation scheme led to instabilities. In particular when parameterizing the prior as a MoG prior (4) these instabilities manifested as exploding prior log-variances (see Fig. 3a) that became evident as the real data distribution became more complex (e.g. CIFAR-10 images vs 2D data). We attribute the aforementioned behavior to the interplay of three aspects: (i) the encoder pushing to suboptimal ELBOs (i.e., suboptimal reconstruction and KL losses) for those samples whose likelihood in fake data distribution is not sufficiently enclosed by the real one (see Proposition 1 and its practical implication in 4.1.2), (ii) hyperparameter-tuning caveats where good results generally required setting the

$\beta_{\mathrm{KL}}$ of the fake ELBO (termed as $\beta_{\mathrm{neg}}$) to be an order of magnitude of the latent dimension (Daniel & Tamar, 2021) and (iii) the behavior of the target distribution in KL minimization where the target variance increases when the source posterior is unlikely under the target distribution (see C.4). Notably, (i) and (ii) promote the posterior of the real samples that overlap with insufficiently enclosed fake ones to diverge from the prior whereas (iii) increases the variance of the prior in an attempt to support a diverging aggregated posterior, which can lead to exploding log-variance in severe cases of (i) – corresponding to the second row of (9). Eliminating (i) or (ii) requires extensive hyper-parameter tuning for each $p_{\mathrm{data}}$, assuming that such a hyper-parameter set even exists. Instead, we opted to address the issue of exploding log-variances by tackling (iii). Namely, we employed an adapting soft-clipping scheme inspired by Chua et al. (2018); Chang et al. (2023) where instabilities were also observed when learning log-variances. Concretely, for each latent dimension $j$, the prior log-variance is clipped to the range $[a_j, b_j]$ using the function $f_c$, defined as:

$$f_c(x) = x + \frac{1}{\beta_j} \cdot \log \frac{1 + \exp(\beta_j \cdot (a_j - x))}{1 + \exp(\beta_j \cdot (x - b_j))}, \tag{11}$$

with $\beta_j = \frac{K}{b_j - a_j}$ and $K$ a positive hyperparameter. The formulation above allows for controlling the steepness of clipping in a unified way using a single hyperparameter $K$ for all latent dimensions. We elaborate further on this choice in C.1. For our all our experiments we used $K = 10$.

### 4.2.3 Responsibilities regularization

Due to the nature of the $L_q$ objective inducing the encoder to act as a discriminator between real and fake data, it is evident that the posterior can diverge arbitrarily from the prior (see Proposition 1).

In practice, we observed that such behavior can cause certain prior components, of a MoG prior, to become more dominant than others in terms of the responsibilities of prior modes to posterior, leading to the formation of inactive prior modes and vanishing gradients (see C.3.2). Consequently, as the aggregated posterior is only supported by a portion of the prior modes, there are not multiple real samples competing for the same region of the latent space leading to unconstrainted generation when sampling for those inactive prior modes. Note that the issue of inactive prior modes formation is applicable both when having a trainable (prior–decoder cooperation) and fixed MoG prior. To alleviate this we employ an entropy regularization on the responsibilities of each prior component discouraging inactive modes from forming. Concretely, the responsibility $c_i$ corresponding to the $i^{\mathrm{th}}$ mode is computed as:

$$c_i = \mathbb{E}_{x \sim p_{\mathrm{data}}(x)} \mathbb{E}_{z \sim q_\phi(z|x)} \left[ \frac{w_i \cdot \mathcal{N}(z|\mu_i, \sigma_i^2 I)}{\sum_{l=1}^{M} w_l \cdot \mathcal{N}(z|\mu_l, \sigma_l^2 I)} \right]. \tag{12}$$

Finally, we define the responsibility vector as:

$$C = [c_1, c_2, \ldots, c_M], \tag{13}$$

and compute its normalized entropy [1] $\mathbb{H}_n(C)$. The $\mathbb{H}_n(C)$ weighted by a non-negative hyperparameter $r_{\mathrm{entropy}}$, is added to the $L_q$ objective. Notably, our responsibility regularization is closely related to the mean entropy maximization regularizer used by Assran et al. (2022); Joulin & Bach (2012) regularizing the mode assignments instead of cluster assignments. Ultimately, encouraging uniform responsibilities accounts for the vanishing gradient issue. We provide the derivation for the prior mode responsibilities in C.3. Fig 3b illustrates a representative case of responsibility entropy development when left unregularized. Note that the responsibility regularization is relevant only for multi-modal priors and therefore not needed in the original S-IntroVAE under the standard Gaussian prior.

---

[1] The entropy is normalized by dividing it by the maximum entropy given $M$ possible assignments, where $M$ is the number of prior components in the MoG.

---

**Algorithm 1** Prior Learning in S-IntroVAE (Daniel & Tamar, 2021). The red-highlighted segments indicate the parts that differ from the standard Gaussian S-IntroVAE. The $L_{\text{rec}}$ and the $L_{\text{KL}}$ refer to the reconstruction loss and the KL divergence between the posterior and the prior target respectively, whereas the $L_{\text{KL}}^{\text{sg}}$ is a modified KL divergence that applies the stop-gradient $\texttt{sg}$ operator on the prior as target.

---

**Require:** $\beta_{rec}, \beta_{\text{KL}}, \beta_{neg}, \gamma, \eta, r_{\text{entropy, K}}$
1: $\phi_E, p_\Lambda, \theta_D \leftarrow$ Initialize network parameters
2: $s \leftarrow 1/$input dim  $\qquad\qquad\qquad\qquad\qquad\qquad\qquad\qquad\qquad$ ▷ Scaling constant
3: $\gamma_r \leftarrow 10^{-8}$  $\qquad\qquad\qquad\qquad\qquad\qquad$ ▷ Scaling parameter for fake data reconstruction
4: $a, b \leftarrow$ Clipping ranged found after the VAE training stage  $\qquad$ ▷ A VAE training stage precedes adversarial training

5: **while** not converged **do**
6: $\qquad x_{\text{real}} \leftarrow$ Random mini-batch from dataset
7: $\qquad z_\mu, z_{\text{logvar}}, w \leftarrow$ Get MoG prior parameters from $p_\Lambda$
8: $\qquad z_{\text{logvar}}^C \leftarrow \textsc{ClipLogvariance}(z_{\text{logvar}}, a, b, K)$
9: $\qquad z_s \leftarrow \textsc{SampleFromMoG}(z_\mu, z_{\text{logvar}}^C, w)$

10: $\qquad \textsc{UpdateEncoder}(x_{\text{real}}, z_s, \phi_E,\, \beta_{rec}, \beta_{\text{KL}}, \beta_{neg},\, r_{\text{entropy}},\, \eta)$
11: $\qquad \textsc{UpdatePriorAndDecoder}(x_{\text{real}}, z_s, p_\Lambda,\, \theta_D, \beta_{rec}, \beta_{\text{KL}}, \gamma, \gamma_r, \eta)$
12: **end while**

13: **procedure** $\textsc{UpdateEncoder}(x_{\text{real}}, z_s, \phi_E,\, \beta_{rec}, \beta_{\text{KL}}, \beta_{neg},\, r_{\text{entropy}},\, \eta)$
14: $\qquad W \leftarrow -s \cdot (\beta_{\text{rec}} \cdot L_{\text{rec}}(x_{\text{real}}) + \beta_{\text{KL}} \cdot L_{\text{KL}}(x_{\text{real}}))$
15: $\qquad W_f \leftarrow -s \cdot (\beta_{\text{rec}} \cdot L_{\text{rec}}(D(z_s)) + \beta_{\text{neg}} \cdot L_{\text{KL}}(D(z_s)))$
16: $\qquad \exp W_f \leftarrow 0.5 \cdot \exp(2 \cdot W_f)$
17: $\qquad C = \textsc{ComputeResponsibilities}(x_{\text{real}})$
18: $\qquad \text{Entropy}_C = \textsc{NormalizedEntropy}(C)$
19: $\qquad L_E \leftarrow W - \exp W_f + s \cdot r_{\text{entropy}} \cdot \text{Entropy}_C$
20: $\qquad \phi_E \leftarrow \phi_E + \eta \nabla_{\phi_E}(L_E)$  $\qquad\qquad\qquad\qquad\qquad\qquad\qquad$ ▷ Adam update
21: **end procedure**

22: **procedure** $\textsc{UpdatePriorAndDecoder}(x_{\text{real}}, z_s, p_\Lambda, \theta_D, \beta_{rec}, \beta_{\text{KL}}, \gamma, \gamma_r, \eta)$
23: $\qquad W \leftarrow -s \cdot (\beta_{\text{rec}} \cdot L_{\text{rec}}(x_{\text{real}}) + \beta_{\text{KL}} \cdot L_{\text{KL}}(x_{\text{real}}))$
24: $\qquad W_f \leftarrow -s \cdot (\gamma_r \cdot \beta_{\text{rec}} \cdot L_{\text{rec}}(\texttt{sg}(D(z_s))) + \beta_{\text{KL}} \cdot L_{\text{KL}}^{\text{sg}}(D(z_s)))$
25: $\qquad L_{PD} \leftarrow W + \gamma \cdot W_f$
26: $\qquad \theta_D \leftarrow \theta_D + \eta \cdot \nabla_{\theta_D}(L_{PD})$  $\qquad\qquad\qquad\qquad\qquad\qquad$ ▷ Adam update
27: $\qquad p_\Lambda \leftarrow p_\Lambda + \eta \cdot \nabla_{p_\Lambda}(L_{PD})$  $\qquad\qquad\qquad\qquad\qquad\qquad$ ▷ Adam update
28: **end procedure**

29: **function** $\textsc{ClipLogvariance}(z_{\text{logvar}}, a, b, K)$
30: $\qquad z_{\text{logvar}}^C \leftarrow$ Clipping the log-variance  $\qquad\qquad\qquad\qquad\qquad\qquad$ ▷ Eq. 11
31: $\qquad$ **return** $z_{\text{logvar}}^C$
32: **end function**

33: **function** $\textsc{SampleFromMoG}(z_\mu, z_{\text{logvar}}, w)$
34: $\qquad i \leftarrow$ Samples a mode index from Categorical($w$)
35: $\qquad z_{\text{std}}^{(i)} \leftarrow \exp\left(0.5 \cdot z_{\text{logvar}}^{(i)}\right)$
36: $\qquad z_s \leftarrow$ Samples from $\mathcal{N}(z_\mu^{(i)}, z_{\text{std}}^{(i)})$
37: $\qquad$ **return** $z_s$
38: **end function**

39: **function** $\textsc{ComputeResponsibilities}(x)$
40: $\qquad$ Compute the expected responsibilities for each mixture component  $\qquad\qquad$ ▷ Eq. 12
41: $\qquad$ Construct the responsibility vector $C$  $\qquad\qquad\qquad\qquad\qquad\qquad$ ▷ Eq. 13
42: $\qquad$ **return** $C$
43: **end function**

44: **function** $\textsc{NormalizedEntropy}(C)$
45: $\qquad$ Compute the entropy of responsibility vector C
46: $\qquad$ Normalize the entropy  $\qquad\qquad\qquad\qquad\qquad\qquad\qquad\qquad$ ▷ Footnote 1
47: $\qquad$ **return** $\text{Entropy}_C$
48: **end function**

---

# 5 Experiments

In this section we investigate the impact of learning the prior in S-IntroVAE. Our testbed consists of a 2D density estimation benchmark alongside three image datasets of varying complexity. To crystallize the effect of prior learning we compare multiple key prior configurations with varying levels of flexibility. Namely, we considered the standard Gaussian, the fixed multi-modal MoG and the trainable multi-modal MoG priors while also ablate over learnable and uniform mixture contributions, when relevant. Concretely the prior configurations considered in our experiments are:

- **Standard Gaussian:** The commonly used isotropic Gaussian prior $\mathcal{N}(0, I)$.

- **Fixed MoG with uniform component contributions:** A VampPrior with uniform contribution weights that is trained during the VAE stage, turned into a MoG [2] and remained fixed during the adversarial training.

- **Trainable MoG with uniform component contributions:** A VampPrior with uniform contribution weights that is trained during the VAE stage, turned into a MoG and continue being trained throughout the adversarial training.

- **Fixed MoG with learnable component contributions:** A VampPrior with learnable component contributions weights that is trained during the VAE stage, turned into a MoG and remained fixed during the adversarial training.

- **Trainable MoG with learnable component contributions:** A VampPrior with learnable component contributions weights that is trained during the VAE stage, turned into a MoG and continue being trained throughout the adversarial training.

Importantly, any argument in favor of prior learning (i.e., trainable MoG) should be supported by performance improvements over both the standard Gaussian and the fixed MoG configurations.

## 5.1 2D - density estimation

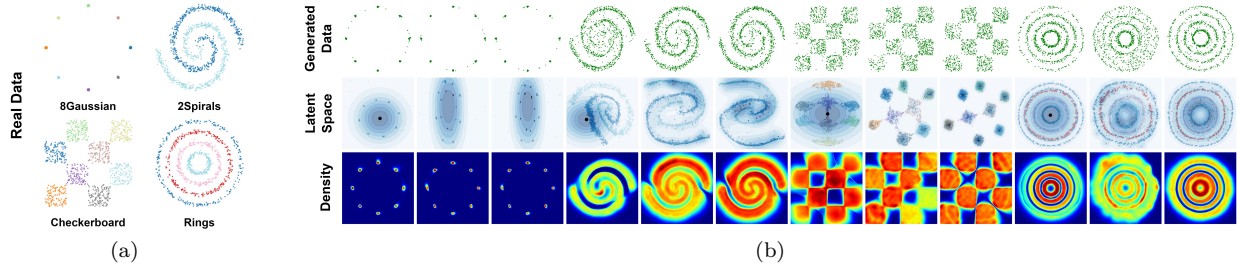

(a)                                       (b)

Figure 4: (a) Real data (b) Qualitative results on density estimation, within each dataset we provide, from left to right, the results under the standard Gaussian, fixed and trainable MoG with learnable contributions corresponding to the $2^{\text{nd}}$, $5^{\text{th}}$ and $6^{\text{th}}$ columns in Table 1 respectively.

For the Density estimation benchmark, we adopt the same evaluation scheme as originally used in S-IntroVAE (Daniel & Tamar, 2021), namely, we use the gnELBO (grid-normalized ELBO) and the histogram-based KL and JSD (Jensen–Shannon divergence) divergences as measures of the inference, the forward and reverse generation capabilities, respectively.

To understand how modeling the prior as a third player affects S-IntroVAE we compare three discrete prior settings, namely (i) standard Gaussian, (ii) fixed MoG and (iii) trainable MoG in decoder-cooperation,

---

[2]The VampPrior was turned into a MoG by loading the aggregated posterior of the pseudoinputs into the parameters of a MoG, ultimately breaking the prior-encoder pairing.

termed as Intro-Prior (IP) (see Fig. 1 for a conceptual visualization of the three settings). When utilizing MoG priors we experimented with both uniform and learnable contribution (LC) of each mode while we modeled the multi-modal prior using 64 components. More specifically, for the LC configuration, the contributions were learnable for both (ii) and (iii) during the VAE pre-training stage whereas during the adversarial training remained learnable only for (iii). The VampPrior was used during the VAE stage due to its benefits in latent space structuring over the MoG (Tomczak & Welling, 2018). The latter was turned into a MoG (see Footnote 2) during the adversarial training to ensure prior–decoder cooperation and to exploit the properties of its NE as given by Corollary 1. More specifically, we note that under the VampPrior, the prior is paired with the encoder establishing a prior–encoder cooperation. As analyzed in B.2.1, this cooperation leads to the prior and the decoder pulling the generated data distribution toward potentially incompatible objectives. When it comes to the regularization, the beta-adapting log-variance clipping was used for IP with $K = 10$ and $[a_j, b_j]$ set to the minimum and maximum log-variance in each latent dimension $j$ as found during the VAE warm-up while the mode responsibilities were left unregularized (i.e., $r_{\text{entropy}} = 0$). For all prior settings, we used 100 Monte Carlo samples to approximate the KL divergence between uni- and multi-modal Gaussian distributions.

In line with Daniel & Tamar (2021), we identified the optimal hyperparameters (i.e., $\beta_{\text{rec}}$, $\beta_{\text{KL}}$ and $\beta_{\text{neg}}$) by performing an extensive grid-search while we used $\alpha = 2$ and $\gamma = 1$.

In Table 1 we report the average (mean $\pm$ standard error) performance across five seeds. As already reported by Daniel & Tamar (2021) the VAE formulation lags behind the S-IntroVAE across all metrics. Regarding prior learning in S-IntroVAE, the quantitative results suggest that in most cases, IP improves the generation performance compared to when using the SG prior or the fixed MoG. In particular, this is more evident when looking at the histogram-based KL metric. The observation above aligns with our intuition as according to Corollary 1 both the prior and the decoder players cooperate towards minimizing the $\text{KL}[p_{\text{data}}(x)||p_d^\lambda(x)]$ term boosting the forward generation performance. An exception to this trend is observed on the 8Gaussian dataset, where training under the standard Gaussian prior achieves the best generation results. Since the 8Gaussian displays the most multi-modal structure (i.e., large areas of low density) we attribute this deviation to the trade-off between stability and modeling multi-modal distribution with push-forward models as discussed in (Salmona et al., 2022).

| Model → | VAE | S-IntroVAE | S-IntroVAE | | | |
|---|---|---|---|---|---|---|
| Prior Type → | SG | | MoG(64) | | | |
| LC Flag → | N/A | N/A | ✗ | ✗ | ✓ | ✓ |
| IP Flag → | N/A | N/A | ✗ | ✓ | ✗ | ✓ |
| **8Gaussian** gnELBO ↓ | 7.48 ±0.03 | 0.51 ±0.07 | 3.62 ±0.31 | 4.8 ±0.15 | **0.25** ±0.02 | 0.26 ±0.04 |
| KL ↓ | 6.94 ±0.36 | **1.23** ±0.05 | 4.46 ±2.63 | 2.36 ±0.3 | 1.94 ±0.33 | 2.24 ±0.71 |
| JSD ↓ | 17.41 ±0.12 | **1.01** ±0.08 | 1.77 ±0.7 | 1.79 ±0.09 | 1.13 ±0.12 | 1.08 ±0.07 |
| **2Spirals** gnELBO ↓ | 6.23 ±0.01 | 6.41 ±0.27 | 6.41 ±0.43 | 6.04 ±0.36 | **5.81** ±0.4 | 6.47 ±0.28 |
| KL ↓ | 10.18 ±0.16 | 9.5 ±0.55 | 8.61 ±0.35 | 8.31 ±0.2 | 9.45 ±0.56 | **8.02** ±0.11 |
| JSD ↓ | 4.94 ±0.11 | 4.21 ±0.22 | 3.76 ±0.04 | **3.53** ±0.04 | 3.89 ±0.08 | 3.64 ±0.07 |
| **Checkerboard** gnELBO ↓ | 8.62 ±0.05 | **7.21** ±0.05 | 8 ±0.03 | 7.66 ±0.13 | 7.81 ±0.09 | 7.67 ±0.06 |
| KL ↓ | 20.79 ±0.08 | 19.62 ±0.25 | 18.58 ±0.22 | 18.72 ±0.27 | 19.04 ±0.65 | **17.7** ±0.11 |
| JSD ↓ | 9.97 ±0.06 | 8.87 ±0.07 | 8.65 ±0.04 | 8.71 ±0.08 | 8.9 ±0.22 | **8.46** ±0.07 |
| **Rings** gnELBO ↓ | 6.37 ±0.04 | **6.03** ±0.05 | 6.73 ±0.18 | 6.86 ±0.16 | 6.4 ±0.34 | 6.65 ±0.18 |
| KL ↓ | 13.3 ±0.28 | 9.99 ±0.27 | 10.07 ±0.37 | **9.77** ±0.31 | 11.31 ±0.56 | 10.31 ±1.03 |
| JSD ↓ | 7.4 ±0.08 | **4.05** ±0.07 | 4.13 ±0.08 | 4.12 ±0.07 | 5.19 ±0.33 | 4.33 ±0.33 |

Table 1: Quantitative performance on the four 2D datasets was evaluated. The LC flag refers to the component contributions being learnable while the IP flag refers to training the prior (i.e., prior–decoder cooperation scheme). Reported values are mean $\pm$ standard error over five runs.

Additionally, when evaluating the qualitative performance as depicted in Fig. 4 we observe that the IP formulation tends to give rise to better-separated clusters in the latent space, more intuitive support of the aggregated posterior, and fewer samples in between the modes.

## 5.2 Image generation

We investigate whether and to which extent prior learning improves the generation performance and the representation learned using the (F)-MNIST and CIFAR-10 datasets. We evaluate the generation quality using the FID metric for samples generated from sampling from the prior and the aggregated posterior denoted as FID(GEN) and FID(REC) respectively. To get a better, more holistic view of how prior learning impacts the generation, we also report the recall and precision metrics (Kynkäänniemi et al., 2019), denoted as Recall(GEN) and precision(GEN) respectively.

The quality of the representations learned by the encoder was evaluated by fitting a linear SVM, similar to Kviman et al. (2023), using 2K-SVM and 10K-SVM iterations as well as utilizing a k-nearest neighbor classifier (k-NN) using 5-NN or 100-NN (Caron et al., 2021).

We use the default training hyperparameters and architectures as provided by Daniel & Tamar (2021) to train the S-IntroVAE, except that the first 20 epochs were used as a VAE training warm-up. We conduct experiments using the same configuration used for the 2D data, while we employ the $r_{\text{entropy}}$ regularization with a value chosen from $\{0, 1, 10, 100\}$ and report the quantitative results for the one that led to the optimal FID(GEN) for each prior setting. The prior was modeled using 10 and 100 components and found that the latter is superior across all metrics, whether using the fixed or trainable MoG configurations, which is an indication that using a sufficiently large number of components is essential. The results provided in Table 2 suggest that replacing the standard Gaussian with a MoG prior (either fixed or trainable) can benefit both the quality of the generation and the learned representation, however, the benefit is less profound in CIFAR-10 compared to the (F)-MNIST datasets. We attribute this behavior to CIFAR-10 potentially being (close-to) uni-modal distribution (Salmona et al., 2022) as opposed to (F)-MNIST which are more likely to be multi-modal.

At this stage, it is natural to question whether the increased generation performance of the MoG configurations is a byproduct of memorized samples in the mixture modes. In this regard, a high precision accompanied with low recall would be an indication of model memorizing specific training samples at the expense of distribution coverage. The results shown in Table 2 do not hint such sample memorization behavior, that is, the relative relationship between recall and precision is similar across all settings.

When comparing fixed (w/o IP) to trainable (w/ IP) MoG priors, we observe a trend where the IP achieves optimal generation performance in two out of the three image benchmarks. When it comes to linear separability, the IP significantly improves over the fixed MoG for two out of the three benchmarks. Interestingly, learning the prior significantly improves the classification performance under the k-NN model across all datasets. This suggests that prior learning in S-IntroVAE leads to a more defined class separation and more interpretable latent space, where similar samples are more effectively clustered together.

A qualitative inspection of the latent space (see Fig. 5) reveals that modeling the prior as a mixture of MoG results in better-separated clusters compared to a standard Gaussian. When comparing a fixed MoG to a trainable MoG, the improvement in class separation is less pronounced but still noticeable which aligns with the quantitative results shown in Table 2. For the complete results and latent space visualization, we refer the readers to D.2 and D.4 respectively.

Finally, it is worth noting how the entropy regularization behaves differently based on the training hyperparameters, dataset complexity and prior learning configuration. In this regard, we observe that a higher $r_{\text{entropy}}$ was nec-

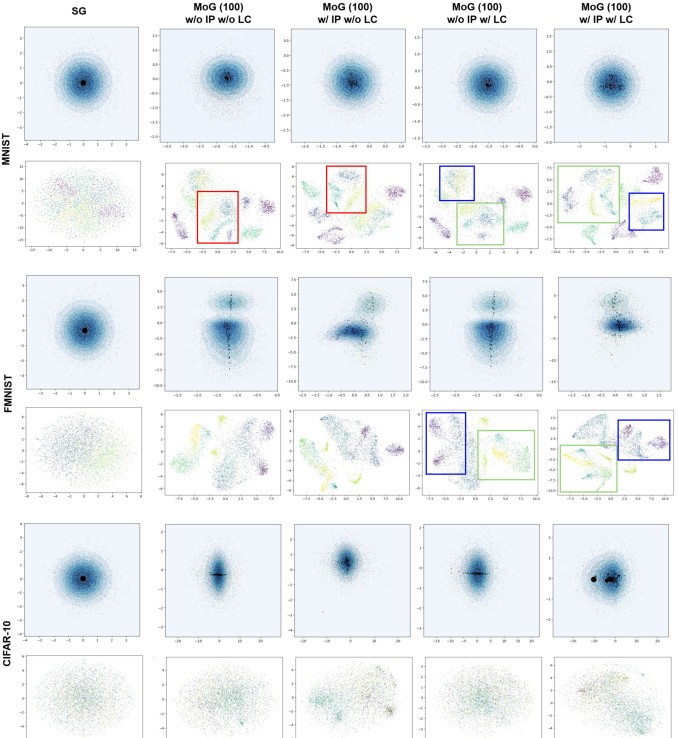

Figure 5: Visualizing the first two latent dimensions of the latent space and the t-SNE 2D embeddings of the full latent space. The columns correspond to those in Tab. 2. Different colors correspond to different classes. The black dots refer to the means of the prior components and their size corresponds to their contribution weight.

essary to achieve the optimal performance on CIFAR-10 compared to the (F)-MNIST datasets under the IP

| | Model → | S-IntroVAE | S-IntroVAE | | | |
|---|---|---|---|---|---|---|
| | Prior Type → | SG | MoG(100) | | | |
| | LC Flag → | N/A | ✗ | ✗ | ✓ | ✓ |
| | IP Flag → | N/A | ✗ | ✓ | ✗ | ✓ |
| **MNIST** | $r_{\text{entropy}}$ | 0 | 10 | 10 | 1 | 10 |
| | Entr. | 0 | $0.892_{\pm0.002}$ | $0.882_{\pm0.001}$ | $0.882_{\pm0.002}$ | $0.853_{\pm0.004}$ |
| | FID (GEN) ↓ | $1.414_{\pm0.025}$ | $1.322_{\pm0.025}$ | $1.352_{\pm0.052}$ | $1.32_{\pm0.061}$ | $\mathbf{1.309}_{\pm\mathbf{0.027}}$ |
| | FID (REC) ↓ | $1.503_{\pm0.031}$ | $\mathbf{1.342}_{\pm\mathbf{0.05}}$ | $1.473_{\pm0.1}$ | $1.363_{\pm0.075}$ | $1.385_{\pm0.081}$ |
| | Recall (GEN)↑ | $\mathbf{0.565}_{\pm\mathbf{0.003}}$ | $0.562_{\pm0.003}$ | $0.553_{\pm0.008}$ | $0.556_{\pm0.003}$ | $0.557_{\pm0.001}$ |
| | Precision (GEN)↑ | $0.522_{\pm0.004}$ | $0.55_{\pm0.005}$ | $0.556_{\pm0.001}$ | $0.561_{\pm0.005}$ | $\mathbf{0.562}_{\pm\mathbf{0.005}}$ |
| | 2K-SVM↑ | $0.93_{\pm0.001}$ | $0.961_{\pm0.001}$ | $0.97_{\pm0.004}$ | $0.962_{\pm0.002}$ | $\mathbf{0.972}_{\pm\mathbf{0.002}}$ |
| | 10K-SVM↑ | $0.93_{\pm0.001}$ | $0.961_{\pm0.001}$ | $0.97_{\pm0.004}$ | $0.962_{\pm0.002}$ | $\mathbf{0.972}_{\pm\mathbf{0.002}}$ |
| | 5-NN↑ | $0.763_{\pm0.003}$ | $0.916_{\pm0.004}$ | $0.947_{\pm0.011}$ | $0.92_{\pm0.001}$ | $\mathbf{0.957}_{\pm\mathbf{0.004}}$ |
| | 100-NN↑ | $0.87_{\pm0.003}$ | $0.934_{\pm0.002}$ | $0.953_{\pm0.007}$ | $0.935_{\pm0.001}$ | $\mathbf{0.958}_{\pm\mathbf{0.002}}$ |
| **FMNIST** | $r_{\text{entropy}}$ | 0 | 0 | 10 | 10 | 10 |
| | Entr. | 0 | $0.931_{\pm0.003}$ | $0.931_{\pm0.001}$ | $0.944_{\pm0.001}$ | $0.903_{\pm0.005}$ |
| | FID (GEN) ↓ | $3.326_{\pm0.039}$ | $2.785_{\pm0.051}$ | $3.025_{\pm0.139}$ | $\mathbf{2.727}_{\pm\mathbf{0.079}}$ | $2.831_{\pm0.1}$ |
| | FID (REC) ↓ | $3.76_{\pm0.097}$ | $\mathbf{2.994}_{\pm\mathbf{0.05}}$ | $3.129_{\pm0.095}$ | $3.185_{\pm0.101}$ | $3.511_{\pm0.074}$ |
| | Recall (GEN)↑ | $0.314_{\pm0.012}$ | $\mathbf{0.35}_{\pm\mathbf{0.003}}$ | $0.336_{\pm0.007}$ | $0.346_{\pm0.004}$ | $0.341_{\pm0.008}$ |
| | Precision (GEN)↑ | $0.518_{\pm0.009}$ | $0.553_{\pm0.005}$ | $0.558_{\pm0.004}$ | $\mathbf{0.576}_{\pm\mathbf{0.006}}$ | $0.574_{\pm0.003}$ |
| | 2K-SVM↑ | $0.681_{\pm0.001}$ | $\mathbf{0.731}_{\pm\mathbf{0.003}}$ | $0.695_{\pm0.007}$ | $0.712_{\pm0.005}$ | $0.696_{\pm0.003}$ |
| | 10K-SVM↑ | $0.731_{\pm0.006}$ | $\mathbf{0.78}_{\pm\mathbf{0.002}}$ | $0.772_{\pm0.003}$ | $0.778_{\pm0.002}$ | $0.773_{\pm0.002}$ |
| | 5-NN↑ | $0.425_{\pm0.009}$ | $0.683_{\pm0.006}$ | $0.693_{\pm0.008}$ | $0.678_{\pm0.006}$ | $\mathbf{0.707}_{\pm\mathbf{0.005}}$ |
| | 100-NN↑ | $0.606_{\pm0.014}$ | $0.736_{\pm0.003}$ | $0.729_{\pm0.006}$ | $0.731_{\pm0.003}$ | $\mathbf{0.739}_{\pm\mathbf{0.004}}$ |
| **CIFAR-10** | $r_{\text{entropy}}$ | 0 | 10 | 100 | 100 | 10 |
| | Entr. | 0 | $0.839_{\pm0.007}$ | $0.94_{\pm0.002}$ | $0.929_{\pm0.003}$ | $0.511_{\pm0.043}$ |
| | FID (GEN) ↓ | $4.424_{\pm0.064}$ | $4.465_{\pm0.038}$ | $\mathbf{4.385}_{\pm\mathbf{0.140}}$ | $4.417_{\pm0.031}$ | $4.594_{\pm0.235}$ |
| | FID (REC) ↓ | $4.13_{\pm0.068}$ | $4.205_{\pm0.091}$ | $\mathbf{4.084}_{\pm\mathbf{0.006}}$ | $4.141_{\pm0.039}$ | $4.585_{\pm0.373}$ |
| | Recall (GEN)↑ | $\mathbf{0.283}_{\pm\mathbf{0.003}}$ | $0.281_{\pm0.001}$ | $\mathbf{0.283}_{\pm\mathbf{0.003}}$ | $0.282_{\pm0.008}$ | $0.264_{\pm0.012}$ |
| | Precision (GEN)↑ | $\mathbf{0.685}_{\pm\mathbf{0.004}}$ | $0.676_{\pm0.002}$ | $0.679_{\pm0.004}$ | $0.677_{\pm0.007}$ | $\mathbf{0.685}_{\pm\mathbf{0.006}}$ |
| | 2K-SVM↑ | $0.245_{\pm0.009}$ | $0.25_{\pm0.002}$ | $\mathbf{0.271}_{\pm\mathbf{0.006}}$ | $0.26_{\pm0.002}$ | $0.256_{\pm0.003}$ |
| | 10K-SVM↑ | $0.391_{\pm0.005}$ | $0.396_{\pm0.003}$ | $\mathbf{0.407}_{\pm\mathbf{0.007}}$ | $0.401_{\pm0.002}$ | $0.396_{\pm0.002}$ |
| | 5-NN↑ | $0.206_{\pm0.001}$ | $0.189_{\pm0}$ | $\mathbf{0.239}_{\pm\mathbf{0.005}}$ | $0.196_{\pm0.001}$ | $0.219_{\pm0.002}$ |
| | 100-NN↑ | $0.308_{\pm0.007}$ | $0.216_{\pm0.008}$ | $\mathbf{0.32}_{\pm\mathbf{0.005}}$ | $0.259_{\pm0.003}$ | $0.273_{\pm0.004}$ |

Table 2: Quantitative performance on the images datasets. The LC flag refers to the component contributions being learnable while the IP flag refers to training the prior (i.e., prior–decoder cooperation scheme). Reported values are mean ± standard error over three runs. The $r_{\text{entropy}}$ row corresponds to the regularization used to obtain the optimal FID(GEN) for each training configuration, where the Entr. row refers to the normalized entropy of the responsibilities where the closer to one its value the more uniformly the aggregated posterior is supported by the prior components.

configuration. Additionally, allowing for learnable contributions under the IP configuration tends to decrease the normalized entropy of the responsibilities suggesting that contributions tend to vanish as soon as they no longer support the aggregated posterior which advocates for the importance of taking measures (e.g., using

the $r_{\text{entropy}}$) to utilize all the components when performing the discrimination (i.e., updating the encoder). For the full ablation on the $r_{\text{entropy}}$ parameter we refer readers to D.3.

## 6 Conclusions

In this study, we have proposed a prior–decoder cooperation scheme as a theoretically sound approach to prior learning in S-IntroVAE, marking the first successful integration of prior learning in Introspective VAEs. Our approach aims to combine two independent directions for improving VAEs: prior learning and the incorporation of adversarial objectives. To realize our proposed scheme, we identified several challenges, which we addressed with theoretically motivated regularization techniques, specifically (i) adaptive log-variance clipping and (ii) responsibility regularization. Our experimental results conducted on 2D and high-dimensional image settings demonstrate the effects of learning the prior in S-IntroVAE. These include a better-structured and more explainable latent space and, in most cases, improved generation performance. We firmly believe that our theoretical insights, coupled with the empirical results, pave the way towards a better understanding of Introspective VAEs and their connection to their VAEs and GANs counterparts. Finally, owing to the unique nature of the problem where a multimodal distribution constitutes both the source and the target, we hope that our analyses enjoy practical use in other areas that deal with problems of similar characteristics e.g., Idempotent Generative Networks (Shocher et al., 2023) or adversarially robust clustering (Yang et al., 2020).

## Acknowledgements

We thank Emanuel Sanchez Aimar and Shashi Nagarajan for the discussions during the early stage of the study. This work was supported by the Wallenberg Artificial Intelligence, Autonomous Systems and Software Program (WASP), funded by the Knut and Alice Wallenberg Foundation. The computational resources were provided by the National Academic Infrastructure for Supercomputing in Sweden (NAISS) at C3SE, partially funded by the Swedish Research Council through grant agreement no. 2022-06725.

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

## A    Preliminaries

The ELBO, given a sample $x$, can be formulated as:

$$
\begin{aligned}
W(x; q, d) &= \mathbb{E}_{z \sim q_{(z|x)}}[\log p_d(x|z)] - \mathrm{KL}[q(z|x)||p(z)] \\
&= \mathbb{E}_{z \sim q_{(z|x)}}[\log p_d(x|z)] - \mathbb{E}_{z \sim q_{(z|x)}}[\log \frac{q(z|x)}{p_z(z)}] \\
&= \mathbb{E}_{z \sim q_{(z|x)}}[\log \frac{p_d(z|x) \cdot p_d(x)}{p_z(z)} - \log \frac{q(z|x)}{p_z(z)}] \\
&= \mathbb{E}_{z \sim q_{(z|x)}}[\log p_d(z|x) + \log p_d(x) - \log q(z|x)] \\
&= \log p_d(x) - \mathrm{KL}[q(z|x)||p_d(z|x)] \leq \log p_d(x),
\end{aligned}
\tag{14}
$$

with $\mathrm{KL}[\cdot||\cdot]$ denoting the Kullback–Leibler (KL) divergence.

## B    Nash Equilibrium in S-IntroVAE

In this section, we provide the theorems based on which the prior–decoder cooperation emerges as a viable option for learning the prior in S-IntroVAE. First, we revisit the derivation of the Nash Equilibrium (NE), under the fixed prior case (originally provided by Daniel & Tamar (2021)), which we modify to account for samples outside the support of the real data distribution. The details and the motivation behind the aforementioned modification are provided in Section B.1.

For simplicity, our analysis is conducted in the discrete domain which is in practice sufficiently revealing as we deal with finite data. From a theoretical standpoint, we can rely on continuity arguments under the assumption of Leibniz's continuity.

### B.1    S-IntroVAE under a fixed prior ((Daniel & Tamar, 2021))

The adversarial game as defined by Daniel & Tamar (2021):

$$
\begin{aligned}
L_q(q, d) &= \mathbb{E}_{p_{\text{data}}}[W(x; q, d)] - \mathbb{E}_{p_d}\left[\frac{1}{\alpha} \cdot \exp(\alpha \cdot W(x; q, d))\right], \\
L_d(q, d) &= \mathbb{E}_{p_{\text{data}}}[W(x; q, d)] + \gamma \cdot \mathbb{E}_{p_d}[W(x; q, d)],
\end{aligned}
\tag{15}
$$

where $\alpha \geq 1$, $\gamma \geq 0$ and $p_d(x) = \mathbb{E}_{p(z)}[p_d(x|z)]$ with $p(z)$ a fixed prior distribution. Note that although originally, a standard Gaussian (SG) prior was used the derivation extends to any prior distribution as long as it is fixed. For notational brevity, we will henceforth refer to the expectation over the real data distribution $\mathbb{E}_{x \sim p_{\text{data}}}[\cdot]$ simply as $\mathbb{E}_{p_{\text{data}}}[\cdot]$, the same applies to the generated data distribution as well.

**Lemma 1.** *Assuming that $[p_d(x)]^{\alpha+1} \leq p_{data}(x)$ for all $x$ such that $p_{data}(x) \geq 0$, the $q^*$ maximizing the $L_q(q, d)$ satisfies $q^*(d)(z|x) = p_d(z|x)$.*

**Remark 2.** *The assumption used in Lemma 1 is a modified version of the one used in (Daniel & Tamar, 2021) in order to account for samples outside of the support of the $p_{data}(x)$. Specifically we require the assumption $[p_d(x)]^{\alpha+1} \leq p_{data}(x)$ to hold for all $x$ such that $p_{data}(x) \geq 0$ instead to $p_{data}(x) > 0$. The utility of this modification is revealed in the proof below.*

*Proof.* Using the ELBO reformulation provided in 14 we develop the $L_q(q, d)$ objective as:

$$
\begin{aligned}
L_q(q,d) &= \mathbb{E}_{p_{\text{data}}}\left[W(x;q,d)\right] - \mathbb{E}_{p_d}\left[\frac{1}{\alpha} \cdot \exp(\alpha \cdot W(x;q,d))\right] \\
&= \mathbb{E}_{p_{\text{data}}}\left[\log p_d(x) - \text{KL}[q(z|x)||p_d(z|x)]\right] \\
&\quad - \frac{1}{a} \cdot \mathbb{E}_{p_d}\left[\exp(\log[p_d(x)]^\alpha - \alpha \cdot \text{KL}[q(z|x)||p_d(z|x)])\right] \\
&= \mathbb{E}_{p_{\text{data}}}\left[\log p_d(x) - \text{KL}[q(z|x)||p_d(z|x)]\right] \\
&\quad - \frac{1}{a} \cdot \mathbb{E}_{p_d}\left[[p_d(x)]^\alpha \cdot \exp(-\alpha \cdot \text{KL}[q(z|x)||p_d(z|x)])\right] \\
&= \sum_x p_{\text{data}}(x) \cdot (\log p_d(x) - \text{KL}[q(z|x)||p_d(z|x)]) - \frac{1}{\alpha} \cdot [p_d(x)]^{\alpha+1} \cdot \exp(-\alpha \cdot \text{KL}[q(z|x)||p_d(z|x)]) \\
&= \begin{cases}
\sum_x p_{\text{data}}(x) \cdot \left(\log p_d(x) - \text{KL}[q(z|x)||p_d(z|x)] - \frac{1}{\alpha} \cdot \frac{[p_d(x)]^{\alpha+1}}{p_{\text{data}}(x)} \cdot \exp(-\alpha \cdot \text{KL}[q(z|x)||p_d(z|x)])\right) \\
\quad = \sum_x G(q,d), \quad x \in \{p_{\text{data}}(x) > 0\} \\
\\
\sum_x \left(-\frac{1}{\alpha} \cdot [p_d(x)]^{\alpha+1} \cdot \exp(-\alpha \cdot \text{KL}[q(z|x)||p_d(z|x)])\right) \\
\quad = \sum_x Q(q,d), \quad x \in \{p_{\text{data}}(x) = 0\}
\end{cases}
\end{aligned}
$$
(16)

The optimal $q^*$ for each $x$ can be found as the maximizer of the $L_q(q,d)$.

Given $x$ such that $p_{\text{data}}(x) > 0$ the optimal $q^*$ can be found as the maximizer of the function $G(q,d)$. In that case, we observe that $q$ contributes to $G(q,d)$ only via the KL term. Based on that, the saddle point can be found by analyzing the derivative of $G(q,d)$ with respect to the KL.

$$
\frac{\partial G(q,d)}{\partial \text{KL}[q(z|x)||p_d(z|x)]} = p_{\text{data}}(x) \cdot \left(-1 + \frac{[p_d(x)]^{a+1}}{p_{\text{data}}(x)} \cdot \exp(-\alpha \cdot \text{KL}[q(z|x)||p_d(z|x)])\right).
$$
(17)

For $x$ such that $p_{\text{data}}(x) > 0$ and $\frac{[p_d(x)]^{\alpha+1}}{p_{\text{data}}(x)} < 1$ , we observe that the $\frac{\partial G(q,d)}{\partial \text{KL}[q(z|x)||p_d(z|x)]} < 0$ for $\text{KL}(q(z|x)||p_d(z|x)) \in [0,\infty)$ (KL is non negative), that is the $G(q,d)$ monotonically decreases with respect to $\text{KL}[q(z|x)||p_d(z|x)]$.

For $x$ such that $p_{\text{data}}(x) > 0$ and $\frac{[p_d(x)]^{\alpha+1}}{p_{\text{data}}(x)} = 1$ we observe that the $\frac{\partial G(q,d)}{\partial \text{KL}[q(z|x)||p_d(z|x)]} = 0$ only when $\text{KL}[q(z|x)||p_d(z|x)] = 0$.
Additionally $\frac{\partial G(q,d)}{\partial^2 \text{KL}[q(z|x)||p_d(z|x)]} = p_{\text{data}}(x) \cdot \left(-\alpha \cdot \frac{[p_d(x)]^{a+1}}{p_{\text{data}}(x)} \cdot \exp(-\alpha \cdot \text{KL}[q(z|x)||p_d(z|x)])\right) \leq 0$.

Based on these two cases above, we conclude that $\text{KL}[q^*(z|x)||p_d(z|x)] = 0$ is a global maxima of $L_q(q,d)$ for $x$ such that $p_{\text{data}}(x) > 0$ and $\frac{[p_d(x)]^{\alpha+1}}{p_{\text{data}}(x)} \leq 1$ .

For $x$ such that $p_{\text{data}}(x) = 0$ the optimal $q^*$ can be found as the maximizer of the function $Q(q,d)$.

$$
\frac{\partial Q(q,d)}{\partial \text{KL}[q(z|x)||p_d(z|x)]} = [p_d(x)]^{a+1} \cdot \exp(-\alpha \cdot \text{KL}[q(z|x)||p_d(z|x)]).
$$
(18)

We observe that $\frac{\partial Q(q,d)}{\partial \text{KL}[q(z|x)||p_d(z|x)]} > 0$, given that $\text{KL}[q(z|x)||p_d(z|x)] \in [0,\infty)$ we conclude $q^*(z|x)$ such that $\text{KL}[q^*(z|x)||p_d(z|x)] = \infty$ is a global maxima of $L_q(q,d)$ for $x$ such that $p_{\text{data}}(x) = 0$. The result above

contradicts what has been argued in (Daniel & Tamar, 2021) and is the motivation behind extending the assumption used in Lemma 1 to account for samples outside of the support of $p_{\text{data}}(x)$ (i.e. $p_{\text{data}}(x) \geq 0$ instead of $p_{\text{data}}(x) > 0$ used in (Daniel & Tamar, 2021)). Under the modified assumption, for x such that $p_{\text{data}}(x) = 0$ we also have $p_d(x) = 0$. In this case samples outside the support of the real data distribution do not contribute to the $L_q(q, d)$ objective and therefore do not influence the optimal $q^*$.

Given that the KL is a proper divergence and under the assumption that $[p_d(x)]^{\alpha+1} \leq p_{\text{data}}(x)$ holds for all $x$ such that $p_{\text{data}}(x) \geq 0$, we conclude that $q^*(z|x) = p_d(z|x))$ is the global maxima of the $L_q(q, d)$, that is:

$$L_q(q(d), d) \leq L_q(q^*(d), d) \text{ for all } q. \tag{19}$$

$\square$

Let us define $d^*$ as:

$$d^* \in \arg\min_d \left\{ \text{KL}[p_{data}(x)||p_d(x)] + \gamma \cdot \mathbb{H}[p_d(x)] \right\}. \tag{20}$$

**Assumption 3** (Modified - (Daniel & Tamar, 2021)). *For all x such that $p_{data}(x) \geq 0$ we have that $[p_{d^*}(x)]^{\alpha+1} \leq p_{data}(x)$.*

**Theorem 2** ((Daniel & Tamar, 2021)). *Under the Assumption 3, the pair of optimal $q^* = p_{d^*}(z|x)$ and $d^*$ as defined in (20) constitutes a NE of the game (15).*

*Proof.* First, we develop the $L_d(q, d)$ as:

$$
\begin{aligned}
L_d(q, d) &= \mathbb{E}_{p_{\text{data}}}\left[W(x; q, d)\right] + \gamma \cdot \mathbb{E}_{p_d}\left[W(x; q, d)\right] \\
&= \mathbb{E}_{p_{\text{data}}}\left[\log p_d(x) - \text{KL}[q(z|x)||p_d(z|x)]\right] \\
&\quad + \gamma \cdot \mathbb{E}_{p_d}\left[\log p_d(x) - \text{KL}[q(z|x)||p_d(z|x)]\right] \\
&= \mathbb{E}_{p_{\text{data}}}\left[\log \frac{p_d(x)}{p_{\text{data}}(x)} + \log p_{\text{data}}(x) - \text{KL}[q(z|x)||p_d(z|x)]\right] \\
&\quad + \gamma \cdot \mathbb{E}_{p_d}\left[\log p_d(x) - \text{KL}[q(z|x)||p_d(z|x)]\right] \\
&= \mathbb{E}_{p_{\text{data}}}[\log p_{\text{data}}(x)] \\
&\quad - \text{KL}[p_{\text{data}}(x)||p_d(x)] - \gamma \cdot \mathbb{H}[p_d(x)] \\
&\quad - \mathbb{E}_{p_{\text{data}}}[\text{KL}[q(z|x)||p_d(z|x)]] - \gamma \cdot \mathbb{E}_{p_d}[\text{KL}[q(z|x)||p_d(z|x)]],
\end{aligned}
\tag{21}
$$

with $\mathbb{H}[\cdot]$ denoting the Shannon entropy. Note that since $\text{KL}[q(z|x)||p_d(z|x)] \geq 0 = \text{KL}[q^*(z|x)||p_d(z|x)]$ the $d^*$ maximizing the $L_d(q, d)$ can be found as the maximizer of $L_d(q^*, d)$. Based on that we set $q = q^*(d)$ in 21 and find the expression of $d$ that maximizes the objective $L_d(q^*(d), d)$ as:

$$
\begin{aligned}
L_d(q^*(d), d) &= \mathbb{E}_{p_{\text{data}}}[\log p_{\text{data}}(x)] \\
&\quad - \text{KL}[p_{\text{data}}(x)||p_d(x)] - \gamma \cdot \mathbb{H}[p_d(x)] \\
&\quad - \underbrace{\mathbb{E}_{p_{\text{data}}}[\text{KL}[q^*(z|x)||p_d(z|x)]]}_{0} - \gamma \cdot \underbrace{\mathbb{E}_{p_d}[\text{KL}[q^*(z|x)||p_d(z|x)]]}_{0},
\end{aligned}
\tag{22}
$$

as the $\mathbb{E}_{p_{\text{data}}}[\log p_{\text{data}}(x)]$ is fixed given a distribution $p_{\text{data}}(x)$ while the $\text{KL}[\cdot||\cdot]$ and $\mathbb{H}[\cdot]$ are non-negative, we can derive the maximizer $d^*$ according to (20). Based on that and according to Lemma 1,

$$
\begin{aligned}
L_q(q(d^*), d^*) &\leq L_q(q^*(d^*), d^*) \text{ for all } q, \\
L_d(q^*(d), d) &\leq L_q(q^*(d^*), d^*) \text{ for all } d,
\end{aligned}
\tag{23}
$$

and therefore we conclude that the pair $q^*$ and $d^*$ such that:

$$
\begin{aligned}
q^*(z|x) &= p_{d^*}(z|x), \\
d^* &\in \arg\min_d \left\{ \mathrm{KL}[p_{\text{data}}(x)||p_d(x)] + \gamma \cdot \mathbb{H}[p_d(x)] \right\}.
\end{aligned}
\tag{24}
$$

is a NE of the (15). $\qquad\qquad\square$

We refer the readers to the original work by Daniel & Tamar (2021) for the proof that for any $p_{\text{data}}(x)$ there always exists $\gamma > 0$ such that the assumption 3 holds for $p_{d^*}(x)$.

## B.2 S-IntroVAE under a trainable prior

Let $\Lambda$ denote the set of possible parameterizations of the prior distributions. We now assume that the prior $p_z(z)$ is learnable and henceforth is denoted as $p_\lambda(z)$ with $\lambda \in \Lambda$ while the generated distribution under that prior is $p_d^\lambda(x) = \mathbb{E}_{p_\lambda(z)} p_d(x|z)$. Consequently the adversarial game (15) is modified as:

$$
\begin{aligned}
L_q(\lambda, q, d) &= \mathbb{E}_{p_{\text{data}}} \left[ W(x; \lambda, q, d) \right] - \mathbb{E}_{p_d^\lambda} \left[ \frac{1}{\alpha} \cdot \exp(\alpha \cdot W(x; \lambda, q, d)) \right], \\
L_d(\lambda, q, d) &= \mathbb{E}_{p_{\text{data}}} \left[ W(x; \lambda, q, d) \right] + \gamma \cdot \mathbb{E}_{p_d^\lambda} \left[ W(x; \lambda, q, d) \right].
\end{aligned}
\tag{25}
$$

### B.2.1 Prior–encoder cooperation

Here we conjecture the infeasibility of learning the prior in collaboration with the encoder while maintaining the same NE of the S-IntroVAE. Intuitively, this formulation seeks to find the optimal prior as the balance between maximizing the real ELBO and minimizing the fake exp(ELBO).

Similarly, the definition in (20) is modified as:

$$
d^*(\lambda) \in \arg\min_d \left\{ \mathrm{KL}[p_{\text{data}}(x)||p_d^\lambda(x)] + \gamma \cdot \mathbb{H}[p_d^\lambda(x)] \right\},
\tag{26}
$$

to account for the parameterized prior. Let $p_d^\lambda(x)$ a discrete distribution of sample size $N$ and $e$'s non-negative real numbers realizing the unnormalized probability masses of $p_d^\lambda(x)$ distribution such that the likelihood of sample $x_k$ is calculated as:

$$
p_d^\lambda(x_k) = \frac{e_k}{\sum\limits_{j=1}^{N} e_j}.
\tag{27}
$$

Let us define the entropy $\mathbb{H}[p_d^\lambda(x)]$ and the $\alpha$-order regularization[3] $\mathbb{A}[p_d^\lambda(x)]$ as:

$$
\mathbb{H}[p_d^\lambda(x)] = -\sum_{i=1}^{N} p_d^\lambda(x_i) \cdot \log\left(p_d^\lambda(x_i)\right),
\tag{28a}
$$

$$
\mathbb{A}[p_d^\lambda(x)] = \sum_{i=1}^{N} p_d^\lambda(x_i) \cdot [p_d^\lambda(x_i)]^\alpha.
\tag{28b}
$$

---

[3]The $\alpha$ hyperparameter is the same used in (15)

**Lemma 2.** *Minimizing $\mathbb{H}[p_d^\lambda(e)]$ with respect to mass $e_k$ requires a positive(negative) update if $\log e_k$ is larger(smaller) than $\mathbb{E}[\log e]$.*

*Proof.* [4] According to the definitions Eqs. (28a) and (27), the entropy can be developed with respect to the probability masses $e$'s as:

$$\mathbb{H}[e] = -\sum_{i=1}^{N} \frac{e_i}{\sum_{j=1}^{N} e_j} \cdot \log\left(\frac{e_i}{\sum_{j=1}^{N} e_j}\right). \tag{29}$$

Based on (29), the derivative of $\mathbb{H}[e]$ with respect to the mass $e_k$ can be computed as:

$$
\begin{aligned}
\frac{\partial \mathbb{H}}{\partial e_k}[e] &= -\frac{\partial}{\partial e_k}\left(\sum_{i=1}^{N} \frac{e_i}{\sum_{j=1}^{N} e_j} \cdot \log\left(\frac{e_i}{\sum_{j=1}^{N} e_j}\right)\right) \\
&= -\frac{\partial}{\partial e_k}\left(\frac{e_k}{\sum_{j=1}^{N} e_j} \cdot \log\left(\frac{e_k}{\sum_{j=1}^{N} e_j}\right) + \sum_{\substack{i=1 \\ i\neq k}}^{N} \frac{e_i}{\sum_{j=1}^{N} e_j} \cdot \log\left(\frac{e_i}{\sum_{j=1}^{N} e_j}\right)\right) \\
&= -\frac{\sum_{j=1}^{N} e_j - e_k}{(\sum_{j=1}^{N} e_j)^2} \cdot \log\left(\frac{e_k}{\sum_{j=1}^{N} e_j}\right) - \frac{\sum_{j=1}^{N} e_j - e_k}{(\sum_{j=1}^{N} e_j)^2} - \sum_{\substack{i=1 \\ i\neq k}}^{N}\left(\frac{e_i}{(\sum_{j=1}^{N} e_j)^2} \cdot \log\left(\frac{e_i}{\sum_{j=1}^{N} e_j}\right) - \frac{e_i}{(\sum_{j=1}^{N} e_j)^2}\right) \\
&= -\frac{\sum_{\substack{j=1 \\ j\neq k}}^{N} e_j}{(\sum_{j=1}^{N} e_j)^2} \cdot \log\left(\frac{e_k}{\sum_{j=1}^{N} e_j}\right) - \frac{\cancel{\sum_{\substack{j=1 \\ j\neq k}}^{N} e_j}}{\cancel{(\sum_{j=1}^{N} e_j)^2}} - \sum_{\substack{i=1 \\ i\neq k}}^{N}\left(\frac{e_i}{(\sum_{j=1}^{N} e_j)^2} \cdot \log\left(\frac{e_i}{\sum_{j=1}^{N} e_j}\right)\right) + \frac{\cancel{\sum_{\substack{i=1 \\ i\neq k}}^{N} e_i}}{\cancel{(\sum_{j=1}^{N} e_j)^2}} \\
&= -\sum_{\substack{i=1 \\ i\neq k}}^{N}\left(\frac{e_i}{(\sum_{j=1}^{N} e_j)^2} \cdot \log\left(\frac{e_k}{e_i}\right)\right) = \sum_{\substack{i=1 \\ i\neq k}}^{N}\left(\frac{e_i}{(\sum_{j=1}^{N} e_j)^2} \cdot \log\left(\frac{e_i}{e_k}\right)\right) \\
&= \sum_{i=1}^{N}\left(\frac{e_i}{(\sum_{j=1}^{N} e_j)^2} \cdot \log\left(\frac{e_i}{e_k}\right)\right) - \cancel{\frac{e_k}{(\sum_{j=1}^{N} e_j)^2} \cdot \log\left(\frac{e_k}{e_k}\right)}^{\,0} \\
&= \frac{1}{\sum_{j=1}^{N} e_j} \cdot \sum_{i=1}^{N}\left(\frac{e_i}{\sum_{j=1}^{N} e_j} \cdot \log\left(\frac{e_i}{e_k}\right)\right) = \frac{1}{\sum_{j=1}^{N} e_j} \cdot \left(\sum_{i=1}^{N}\left(\frac{e_i}{\sum_{j=1}^{N} e_j} \cdot \log e_i\right) - \log e_k\right).
\end{aligned}
\tag{30}
$$

The update towards minimizing the entropy regularization reads as $e_k' = (e_k - \eta \cdot \frac{\partial \mathbb{H}}{\partial e_k}[e])^+$. According to (30), the update $-\eta \cdot \frac{\partial \mathbb{H}}{\partial e_k}[e]$ of mass $e_k$ is positive if $\log e_k$ is larger than $\mathbb{E}[\log e]$ and vice versa. $\qquad\square$

---

[4] The proof was originally provided by Ji et al. (2021)

**Lemma 3.** *Minimizing $\mathbb{A}[p_d^\lambda(e)]$ with respect to mass $e_k$ requires a negative(positive) update if $e_k^\alpha$ is larger(smaller) than $\mathbb{E}[e_k^\alpha]$.*

*Proof.* According to the definitions Eqs. (28b) and (27), the $\alpha$-order regularization can be developed with respect to the probability masses $e$'s as:

$$\mathbb{A}[e] = \sum_{i=1}^{N} \frac{e_i}{\sum_{j=1}^{N} e_j} \cdot \left( \frac{e_i}{\sum_{j=1}^{N} e_j} \right)^\alpha = \sum_{i=1}^{N} \left( \frac{e_i}{\sum_{j=1}^{N} e_j} \right)^{(\alpha+1)}. \tag{31}$$

Based on (31), the derivative of $\mathbb{A}[e]$ with respect to the mass $e_k$ can be computed as:

$$
\begin{aligned}
\frac{\partial \mathbb{A}}{\partial e_k}[e] &= \frac{\partial}{\partial e_k} \left( \sum_{i=1}^{N} \left( \frac{e_i}{\sum_{j=1}^{N} e_j} \right)^{(\alpha+1)} \right) = \frac{\partial}{\partial e_k} \left( \left( \frac{e_k}{\sum_{j=1}^{N} e_j} \right)^{(\alpha+1)} + \sum_{\substack{i=1 \\ i \neq k}}^{N} \left( \frac{e_i}{\sum_{j=1}^{N} e_j} \right)^{(\alpha+1)} \right) \\[2mm]
&= \frac{(\alpha+1) \cdot e_k^\alpha \cdot (\sum_{j=1}^{N} e_j)^{(\alpha+1)}}{(\sum_{j=1}^{N} e_j)^{2 \cdot (\alpha+1)}} - \frac{(\alpha+1) \cdot e_k^{(\alpha+1)} \cdot (\sum_{j=1}^{N} e_j)^\alpha}{(\sum_{j=1}^{N} e_j)^{2 \cdot (\alpha+1)}} - \sum_{\substack{i=1 \\ i \neq k}}^{N} \left( \frac{(\alpha+1) \cdot e_i^{(\alpha+1)} \cdot (\sum_{j=1}^{N} e_j)^\alpha}{(\sum_{j=1}^{N} e_j)^{2 \cdot (\alpha+1)}} \right) \\[2mm]
&= (a+1) \cdot \left( \frac{e_k^\alpha}{(\sum_{j=1}^{N} e_j)^{(\alpha+1)}} - \sum_{i=1}^{N} \left( \frac{e_i^{(\alpha+1)}}{(\sum_{j=1}^{N} e_j)^{(\alpha+2)}} \right) \right) = \frac{(a+1)}{(\sum_{j=1}^{N} e_j)^{(\alpha+1)}} \cdot \left( e_k^\alpha - \sum_{i=1}^{N} \left( \frac{e_i}{\sum_{j=1}^{N} e_j} \cdot e_i^\alpha \right) \right)
\end{aligned} \tag{32}
$$

Similarly to the entropy minimization case, the update towards minimizing the $\alpha$-order regularization reads as $e_k' = (e_k - \eta \cdot \frac{\partial \mathbb{A}}{\partial e_k}[e])^+$. According to (32), the update $-\eta \cdot \frac{\partial \mathbb{A}}{\partial e_k}[e]$ of mass $e_k$ is negative if $e_k^\alpha$ is larger than $\mathbb{E}[e^\alpha]$ and vice versa. $\qquad\square$

$$d^*(\lambda) \in \arg\min_{d} \left\{ \mathrm{KL}[p_{\mathrm{data}}(x)||p_d^\lambda(x)] + \gamma \cdot \mathbb{H}[p_d^\lambda(x)] \right\}, \tag{26 revisited}$$

**Lemma 4.** *For $q^* = p_d^\lambda(z|x)$, the $d^*$ maximizing the $L_d(\lambda, q^*, d)$ satisfies (26).*

*Proof.* Similar to Theorem 2, we develop the $L_d(\lambda, q, d)$ as:

$$
\begin{aligned}
L_d(\lambda, q, d) = {}& \mathbb{E}_{p_{\mathrm{data}}}[\log p_{\mathrm{data}}(x)] \\
& - \mathrm{KL}[p_{\mathrm{data}}(x)||p_d^\lambda(x)] - \gamma \cdot \mathbb{H}[p_d^\lambda(x)] \\
& - \mathbb{E}_{p_{\mathrm{data}}}[\mathrm{KL}[q(z|x)||p_d^\lambda(z|x)]] - \gamma \cdot \mathbb{E}_{p_d^\lambda}[\mathrm{KL}[q(z|x)||p_d^\lambda(z|x)]],
\end{aligned} \tag{33}
$$

with $\mathbb{H}[\cdot]$ denoting the Shannon entropy. Note that since $\mathrm{KL}[q(z|x)||p_d^\lambda(z|x)] \geq 0 = \mathrm{KL}[q^*(z|x)||p_d^\lambda(z|x)]$ the $d^*$ maximizing the $L_d(\lambda, q, d)$ can be found as the maximizer of $L_d(\lambda, q^*, d)$. Based on that we set $q = q^*(\lambda, d)$ in 33 and find the $d^*$ that maximizes the objective $L_d(\lambda, q^*(\lambda, d), d)$ as:

$$L_d(\lambda, q^*(\lambda, d), d) = \mathbb{E}_{p_{\text{data}}}[\log p_{\text{data}}(x)]$$
$$- \text{KL}[p_{\text{data}}(x)||p_d^\lambda(x)] - \gamma \cdot \mathbb{H}[p_d^\lambda(x)]$$
$$- \mathbb{E}_{p_{\text{data}}}[\text{KL}[q^*(z|x)||p_d^\lambda(z|x)]]^{\;\;0} - \gamma \cdot \mathbb{E}_{p_d}[\text{KL}[q^*(z|x)||p_d^\lambda(z|x)]]^{\;\;0}.$$
(34)

Based on (34), we can derive the maximizer $d^*$ according to (26). $\qquad\square$

Let us now define:

$$\lambda^*(d) \in \arg\min_\lambda \left\{ \text{KL}[p_{\text{data}}(x)||p_d^\lambda(x)] + \frac{1}{\alpha} \cdot \mathbb{A}[p_d^\lambda(x)] \right\}.$$
(35)

**Lemma 5.** *Assuming that $[p_d^\lambda(x)]^{\alpha+1} \le p_{data}(x)$ for all $x$ such that $p_{data}(x) \ge 0$, for $q^* = p_d^\lambda(z|x)$, the $\lambda^*$ maximizing the $L_q(\lambda, q^*, d)$ satisfies (35).*

*Proof.* Given the trainable prior $p_\lambda(z)$ the $L_q(\lambda, q, d)$ becomes:

$$L_q(\lambda, q, d) = \sum_x p_{\text{data}}(x)(\log p_d^\lambda(x) - \text{KL}[q(z|x)||p_d^\lambda(z|x)]$$
$$- \frac{1}{\alpha} \cdot [p_d^\lambda(x)]^{\alpha+1} \cdot \exp(-\alpha \cdot \text{KL}[q(z|x)||p_d^\lambda(z|x)])).$$
(36)

Let $q^*(z|x) = p_d^\lambda(z|x)$, the objective $L_q(\lambda, q^*, d)$ reads as:

$$L_q(\lambda, q^*, d) = \sum_x p_{\text{data}}(x) \cdot (\log p_d^\lambda(x) - \text{KL}[q^*(z|x)||p_d^\lambda(z|x)])^{\;\;0}$$

$$- \frac{1}{\alpha} \cdot [p_d^\lambda(x)]^{\alpha+1} \cdot \exp(-\alpha \text{KL}[q^*(z|x)||p_d^\lambda(z|x)])^{\;\;1}$$

$$= \sum_x p_{\text{data}}(x) \cdot \log p_d^\lambda(x) - \frac{1}{\alpha} \cdot [p_d^\lambda(x)]^{\alpha+1}$$

$$= \sum_x p_{\text{data}}(x) \cdot (\log p_d^\lambda(x) - \log p_{\text{data}}(x) + \log p_{\text{data}}(x)) - \frac{1}{\alpha} \cdot [p_d^\lambda(x)]^{\alpha+1}$$

$$= \sum_x p_{\text{data}}(x) \cdot \log \frac{p_d^\lambda(x)}{p_{\text{data}}(x)}$$
(37)

$$+ \sum_x p_{\text{data}}(x) \cdot \log p_{\text{data}}(x) - \frac{1}{\alpha} \cdot \sum_x [p_d^\lambda(x)]^{\alpha+1}$$

$$= -\sum_x p_{\text{data}}(x) \cdot \log \frac{p_{\text{data}}(x)}{p_d^\lambda(x)}$$

$$+ \sum_x p_{\text{data}}(x) \cdot \log p_{\text{data}}(x) - \frac{1}{\alpha} \cdot \sum_x p_d^\lambda(x) \cdot [p_d^\lambda(x)]^{\alpha}$$

$$= -\text{KL}[p_{\text{data}}(x)||p_d^\lambda(x)]$$

$$+ \mathbb{E}_{p_{\text{data}}}[\log p_{\text{data}}(x)] - \frac{1}{\alpha} \cdot \mathbb{A}[p_d^\lambda(x)]$$

Based on (37), we observe that the $\mathbb{E}_{p_{\text{data}}}[\log p_{\text{data}}(x)]$ is fixed given $p_{\text{data}}$ while $\mathbb{A}[\cdot]$ is non-negative, therefore we can derive the maximizer $\lambda^*$ according to 35. $\qquad\square$

$$d^*(\lambda) \in \arg\min_d \left\{ \mathrm{KL}[p_{\mathrm{data}}(x)||p_d^\lambda(x)] + \gamma \cdot \mathbb{H}[p_d^\lambda(x)] \right\}, \qquad \text{(26 revisited)}$$

Lemmas 2 and 3 suggest that minimizing the entropy and the $\alpha$-order push towards the Dirac and uniform distributions respectively. Based on that and the minimization objectives of the $d$ and $\lambda$ players, we formulate a conjecture on the incompatibility of prior–encoder cooperation.

**Conjecture 1.** *When training the prior player $\lambda$ in cooperation with the encoder player $q$ (i.e. to maximize the same $L_q(\lambda, q, d)$ objective), there does not exist $\lambda^*$ such that the triplet $\lambda^*$, $q^*$ satisfiying $q^*(z|x) = p_{d^*}^{\lambda^*}(z|x)$ and $d^*$ as defined in (26) constitutes a NE of the game (25), under the assumption that $p_{d^*}^{\lambda^*}(x,z) \neq p_{d^*}^{\lambda^*}(x) \cdot p_{d^*}^{\lambda^*}(z)$.*

**Remark 3.** *The Conjecture 1 suggests that the prior–encoder cooperation scheme is not a variable option for prior learning in S-IntroVAE, in the sense that it does not share the same NE with its fixed prior counterpart.*

### B.2.2 Prior–decoder cooperation

Here, we consider the same game defined in (25) but under a prior–decoder cooperation scheme where both the prior $\lambda$ and decoder $d$ players maximize the same objective $L_q(\lambda, q, d)$. First, let us extend Lemma 1 for the trainable prior case.

**Lemma 6.** *Assuming that $[p_d^\lambda(x)]^{\alpha+1} \leq p_{data}(x)$ for all $x$ such that $p_{data}(x) \geq 0$, the $q^*$ maximizing the $L_q(\lambda, q, d)$ satisfies $q^*(\lambda, d)(z|x) = p_d^\lambda(z|x)$.*

*Proof.* We develop the $L_q(\lambda, q, d)$ objective as:

$$
\begin{aligned}
L_q(\lambda, q, d) &= \mathbb{E}_{p_{\mathrm{data}}}\left[W(x; \lambda, q, d)\right] - \mathbb{E}_{p_d}\left[\frac{1}{\alpha} \cdot \exp(\alpha \cdot W(x; \lambda, q, d))\right] \\
&= \mathbb{E}_{p_{\mathrm{data}}}\left[\log p_d^\lambda(x) - \mathrm{KL}[q(z|x)||p_d^\lambda(z|x)]\right] \\
&\quad - \frac{1}{a} \cdot \mathbb{E}_{p_d^\lambda}\left[\exp(\log[p_d^\lambda(x)]^\alpha - \alpha \cdot \mathrm{KL}[q(z|x)||p_d^\lambda(z|x)])\right] \\
&= \mathbb{E}_{p_{\mathrm{data}}}\left[\log p_d^\lambda(x) - \mathrm{KL}[q(z|x)||p_d^\lambda(z|x)]\right] \\
&\quad - \frac{1}{a} \cdot \mathbb{E}_{p_d^\lambda}\left[[p_d^\lambda(x)]^\alpha \cdot \exp(-\alpha \cdot \mathrm{KL}[q(z|x)||p_d^\lambda(z|x)])\right] \\
&= \sum_x p_{\mathrm{data}}(x) \cdot (\log p_d^\lambda(x) - \mathrm{KL}[q(z|x)||p_d^\lambda(z|x)]) - \frac{1}{\alpha} \cdot [p_d^\lambda(x)]^{\alpha+1} \cdot \exp(-\alpha \cdot \mathrm{KL}[q(z|x)||p_d^\lambda(z|x)])
\end{aligned}
$$
$$\tag{38}$$

We follow the same reasoning used in Lemma 1 and conclude that under the assumption that $[p_d^\lambda(x)]^{\alpha+1} \leq p_{\mathrm{data}}(x)$ holds for all $x$ such that $p_{\mathrm{data}}(x) \geq 0$ $q^*(z|x) = p_d(z|x))$ is the global maxima of the $L_q(q, d)$, that is:

$$L_q(\lambda, q(\lambda, d), d) \leq L_q(\lambda, q^*(\lambda, d), d) \quad \text{for all } q. \tag{39}$$

$\square$

Let us define $\lambda^*$ and $d^*$ as:

$$(\lambda^*, d^*) \in \arg\min_{\lambda, d} \left\{ \mathrm{KL}[p_{data}(x)||p_d^\lambda(x)] + \gamma \cdot \mathbb{H}[p_d^\lambda(x)] \right\}. \tag{40}$$

Now we also modify the Assumption 3 as:

**Assumption 4.** *For all $x$ such that $p_{data}(x) \geq 0$ we have that $[p_{d^*}^{\lambda^*}(x)]^{\alpha+1} \leq p_{data}(x)$.*

**Corollary 2.** *Under the Assumption 4, when training the prior player $\lambda$ in cooperation with the decoder player $d$ then the triplet $q^* = p_{d^*}^{\lambda^*}(z|x)$, $\lambda^*$ and $d^*$ as defined in (40) constitutes a NE of the game (25).*

*Proof.* Similar to Theorem 2, we develop the $L_d(\lambda, q, d)$ as:

$$
\begin{aligned}
L_d(\lambda, q, d) = {} & \mathbb{E}_{p_{\text{data}}}[\log p_{\text{data}}(x)] \\
& - \text{KL}[p_{\text{data}}(x)||p_d(x)] - \gamma \cdot \mathbb{H}[p_d(x)] \\
& - \mathbb{E}_{p_{\text{data}}}[\text{KL}[q(z|x)||p_d^\lambda(z|x)]] - \gamma \cdot \mathbb{E}_{p_d^\lambda}[\text{KL}[q(z|x)||p_d^\lambda((z|x)]],
\end{aligned}
\tag{41}
$$

with $\mathbb{H}[\cdot]$ denoting the Shannon entropy. Note that since $\text{KL}[q(z|x)||p_d^\lambda(z|x)] \geq 0 = \text{KL}[q^*(z|x)||p_d^\lambda(z|x)]$ the $(\lambda^*, d^*)$ maximizing the $L_d(\lambda, q, d)$ can be found as the maximizer of $L_d(\lambda, q^*, d)$. Based on that we set $q = q^*(\lambda, d)$ in 41 and find the $(\lambda^*, d^*)$ that maximizes the objective $L_d(\lambda, q^*(\lambda, d), d)$ as:

$$
\begin{aligned}
L_d(\lambda, q^*(\lambda, d), d) = {} & \mathbb{E}_{p_{\text{data}}}[\log p_{\text{data}}(x)] \\
& - \text{KL}[p_{\text{data}}(x)||p_d(x)] - \gamma \cdot \mathbb{H}[p_d(x)] \\
& - \mathbb{E}_{p_{\text{data}}}[\underbrace{\text{KL}[q^*(z|x)||p_d(z|x)]}_{0}] - \gamma \cdot \mathbb{E}_{p_d}[\underbrace{\text{KL}[q^*(z|x)||p_d(z|x)]}_{0}].
\end{aligned}
\tag{42}
$$

We can now derive the maximizer $(\lambda^*, d^*)$ according to (40). Based on that and according to Lemma 6,

$$
\begin{aligned}
L_q(\lambda, q(\lambda^*, d^*), d^*) &\leq L_q(q^*(\lambda, d^*), d^*) \ \text{ for all } \ q, \\
L_d(\lambda, q^*(\lambda, d), d) &\leq L_q(q^*(\lambda^*, d^*), d^*) \ \text{ for all } \ \lambda \text{ and } d,
\end{aligned}
\tag{43}
$$

and therefore we conclude that the triplet $\lambda^*$, $q^*$ and $d^*$ such that:

$$
\begin{aligned}
q^*(z|x) &= p_{d^*}^{\lambda^*}(z|x), \\
(\lambda^*, d^*) &\in \arg\min_{\lambda, d} \{\text{KL}[p_{\text{data}}(x)||p_d(x)] + \gamma \cdot \mathbb{H}[p_d(x)]\} .
\end{aligned}
\tag{44}
$$

is a NE of the (25). $\qquad\square$

As the proof of the existence of the $\gamma$ does not assume the nature of the prior, the proof provided by Daniel & Tamar (2021) can be trivially extended for our case of $p_{d^*}^{\lambda^*}$ to show that there exists $\gamma$ such that the $p_{d^*}^{\lambda^*}$ with $(\lambda^*, d^*)$ as defined in (40) satisfies the Assumption 4.

### B.3 Optimal ELBO in the assumption-free setting

In the previous section, the NE of the S-IntroVAE under the prior–decoder cooperation scheme (25) was analyzed under the Assumptions 4. In practice, however, such an assumption might not always be satisfied, particularly in the early stages of training. For instance, it is common in adversarial training for the generator/decoder to generate samples of very low quality (i.e. outside of the support of real data distribution) or to experience mode-collapse (i.e. generating some realistic samples at a disproportionately higher frequency compared to the real data distribution). Evidently, both these cases might lead to violations of said assumption.

Analyzing the behavior of the encoder in the assumption-free setting provides insights into the training dynamics of S-IntroVAE, enabling a better understanding of the method and its relationship to traditional VAEs. Furthermore conducting the analysis with respect to the ELBO $W(x; \lambda, q, d)$ offers a practical tool since the ELBO is comprised of the reconstruction and the KL divergence losses as opposed

to the $\text{KL}[q(z|x)||p_d^\lambda(z|x)]$ term (used in Lemma 6) which is intractable.

Let $\mathbb{X} = \{x | x \in p_{\text{data}}(x) > 0 \cup p_d^\lambda(x) > 0\}$ , we define the ELBO $W(x; \lambda, q^*, d)$ as:

$$W(x; \lambda, q^*, d) = \begin{cases} -\infty, & x \in \{x \in \mathbb{X} \mid p_{\text{data}}(x) = 0\} \\ \frac{1}{\alpha} \cdot \log \frac{p_{\text{data}}(x)}{p_d^\lambda(x)}, & x \in \{x \in \mathbb{X} \mid p_{\text{data}}(x) > 0 \cap [p_d^\lambda(x)]^{\alpha+1} > p_{\text{data}}(x)\} \\ \log p_d^\lambda(x), & x \in \{x \in \mathbb{X} \mid p_{\text{data}}(x) > 0 \cap [p_d^\lambda(x)]^{\alpha+1} \le p_{\text{data}}(x)\}\} \end{cases} \tag{45}$$

**Proposition 2.** *Given a fixed generated data distribution $p_d^\lambda(x)$ the $q^*$ maximizing $L_q(\lambda, d, q)$ in (25) is such that the ELBO $W(x; \lambda, q^*, d)$ satisfies 45.*

*Proof.* Similarly to Lemma 6, we develop $L_q(\lambda, q, d)$ as:

$$L_q(\lambda, q, d) = \sum_x p_{\text{data}}(x) \cdot (\log p_d^\lambda(x) - \text{KL}[q(z|x)||p_d^\lambda(z|x)]) - \frac{1}{\alpha} \cdot [p_d^\lambda(x)]^{\alpha+1} \cdot \exp(-\alpha \cdot \text{KL}[q(z|x)||p_d^\lambda(z|x)])$$

$$= \begin{cases} \sum_x p_{\text{data}}(x) \cdot \left(\log p_d^\lambda(x) - \text{KL}[q(z|x)||p_d^\lambda(z|x)] - \frac{1}{\alpha} \cdot \frac{[p_d^\lambda(x)]^{\alpha+1}}{p_{\text{data}}(x)} \cdot \exp(-\alpha \cdot \text{KL}[q(z|x)||p_d^\lambda(z|x)])\right) \\ = \sum_x G(\lambda, q, d), \quad x \in \{p_{\text{data}}(x) > 0\} \\[1em] \sum_x \left(-\frac{1}{\alpha} \cdot [p_d^\lambda(x)]^{\alpha+1} \exp(-\alpha \cdot \text{KL}[q(z|x)||p_d^\lambda(z|x)])\right) \\ = \sum_x Q(\lambda, q, d), \quad x \in \{p_{\text{data}}(x) = 0\} \end{cases} \tag{46}$$

Again, we can find the $q^*$ maximizing $L_q(\lambda, q, d)$ by analyzing the derivatives of the functions $G(\lambda, q, d)$ and $Q(\lambda, q, d)$. In particular, we identify four cases.

- $x \in \{x \in \mathbb{X} \mid p_{\text{data}}(x) > 0 \cap [p_d^\lambda(x)]^{\alpha+1} > p_{\text{data}}(x)\}$

  In this case, the $q^*$ can be found as:

$$\frac{\partial G(\lambda, q, d)}{\partial \text{KL}[q(z|x)||p_d^\lambda(z|x)]} = 0 \Leftrightarrow$$

$$p_{\text{data}}(x) \cdot \left(-1 + \frac{[p_d^\lambda(x)]^{a+1}}{p_{\text{data}}(x)} \cdot \exp(-\alpha \cdot \text{KL}[q(z|x)||p_d^\lambda(z|x)])\right) = 0 \Leftrightarrow$$

$$\exp(-\alpha \cdot \text{KL}[q(z|x)||p_d^\lambda(z|x)]) = \frac{p_{\text{data}}(x)}{[p_d(x)]^{a+1}} \Leftrightarrow$$

$$-\text{KL}[q(z|x)||p_d^\lambda(z|x)] = \frac{1}{\alpha} \cdot \log \frac{p_{\text{data}}(x)}{[p_d^\lambda(x)]^{a+1}} \Leftrightarrow \tag{47}$$

$$\log p_d^\lambda(x) - \text{KL}[q(z|x)||p_d^\lambda(z|x)] = \frac{1}{\alpha} \cdot \log \frac{p_{\text{data}}(x)}{[p_d^\lambda(x)]^{a+1}} + \log p_d^\lambda(x) \overset{(14)}{\Leftrightarrow}$$

$$W(x; \lambda, q, d) = \frac{1}{\alpha} \cdot \log \frac{p_{\text{data}}(x)}{[p_d^\lambda(x)]^{a+1}} + \frac{1}{\alpha} \cdot \log[p_d^\lambda(x)]^\alpha \Leftrightarrow$$

$$W(x; q, d) = \frac{1}{\alpha} \cdot \log \frac{p_{\text{data}}(x)}{p_d^\lambda(x)}.$$

Note that $\frac{\partial G(\lambda, q, d)}{\partial^2 \text{KL}[q(z|x)||p_d^\lambda(z|x)]} = p_{\text{data}}(x) \cdot \left(-\alpha \cdot \frac{[p_d^\lambda(x)]^{a+1}}{p_{\text{data}}(x)} \cdot \exp(-\alpha \cdot \text{KL}[q(z|x)||p_d^\lambda(z|x)])\right) \le 0$ therefore the $q^*$ such that $W(x; \lambda, q^*, d) = \frac{1}{\alpha} \cdot \log \frac{p_{\text{data}}(x)}{p_d^\lambda(x)}$ is the maximizer of $L_q(\lambda, q, d)$ for $x \in \{x \in \mathbb{X} \mid p_{\text{data}}(x) > 0 \cap [p_d^\lambda(x)]^{\alpha+1} > p_{\text{data}}(x)\}$.

- $x \in \{x \in \mathbb{X} \mid p_{\text{data}}(x) > 0 \cap [p_d^\lambda(x)]^{\alpha+1} \leq p_{\text{data}}(x)\}$

  In this case, the maximizer of $L_q(\lambda, q, d)$ was found in Lemma 6 as the $q^*$ such that $\text{KL}[q^*(z|x)\|p_d^\lambda(z|x)] = 0$. Substracting $\log p_d^\lambda(x)$ to both sides and using (14) we get that the $q^*$ such that $W(x; \lambda, q^*, d) = \log p_d^\lambda(x)$ is the maximizer of $L_q(\lambda, q, d)$ for $x \in \{x \in \mathbb{X} \mid p_{\text{data}}(x) > 0 \cap [p_d^\lambda(x)]^{\alpha+1} \leq p_{\text{data}}(x)\}$.

- $x \in \{x \in \mathbb{X} \mid p_{\text{data}}(x) = 0\}$

  In this case, the maximizer of $L_q(\lambda, q, d)$ was found in Lemma 6 as the $q^*$ such that $\text{KL}[q^*(z|x)\|p_d^\lambda(z|x)] = \infty$. Substracting $\log p_d^\lambda(x)$ to both sides, using (14) and given that $\log p_d^\lambda(x) \leq 0$ we get that the $q^*$ such that $W(x; \lambda, q^*, d) = -\infty$ is the maximizer of $L_q(\lambda, q, d)$ for $x \in \{x \in \mathbb{X} \mid p_{\text{data}}(x) = 0\}$.

- $x \in \{x \in \mathbb{X} \mid p_{\text{data}}(x) = 0 \cap p_d^\lambda(x) = 0\} = \emptyset$

  Note that the $\{p_{\text{data}}(x) = 0 \cap p_d^\lambda(x) = 0\}$ set refers to samples $x$ outside of the support of both real and generated data distributions which are of no practical relevance. In practice, the encoder maximizes the $L_q$ over the expectation of empirical real and generated data distributions, motivating the definition of $\mathbb{X}$ as the union of their supports.

  $\square$

Interestingly, the ELBO $W(x; \lambda, q^*, d)$ at the optimal $q^*$ is a continuous function with respect to $p_{\text{data}}(x)$. Additionally, it is revealed that the higher the sample-wise likelihood mismatch between the real $p_{\text{data}}(x)$ and generated $p_d^\lambda(x)$ data distribution, the lower (more negative) the ELBO $W(x; \lambda, q^*, d)$ is. The aforementioned behavior aligns with our intuition as the encoder in S-IntroVAE acts as a discriminator.

On the other hand, given a fixed $p_d^\lambda(x)$, it can be trivially shown that the encoder of regularly trained VAEs converges to true posterior which is equivalent to $W_{\text{VAE}}{}^5(x; \lambda, q^*, d) = \log p_d^\lambda(x)$. Naturally, these two observations relate the behavior of the encoders of VAEs and S-IntroVAEs where the latter behaves similarly to the former only if $p_d^\lambda(x)$ is sufficiently *enclosed* by the $p_{\text{data}}(x)$. Given a $p_{\text{data}}(x)$, the *enclosed* term refers to the generated data distribution $p_d^\lambda(x)$ for which the Assumption 3 holds.

## C   Implementation

In this section, we provided the details behind some implementation choices.

### C.1   Adaptive variance soft-clipping

The (Chang et al., 2023; Chua et al., 2018) works realize log variance soft-clipping as:

$$
\begin{aligned}
f_c(\text{logvar}) &= \text{logvar} - \text{softplus}(\text{logvar-b}) + \text{softplus}(\text{a - logvar}) \\
&= \text{logvar} - \frac{1}{\beta} \cdot \log(1 + \exp(\beta \cdot (\text{logvar} - b))) \\
&\quad + \frac{1}{\beta} \cdot \log(1 + \exp(\beta \cdot (a - \text{logvar}))),
\end{aligned}
\tag{48}
$$

where $f_c(\text{logvar})$ is the soft-clipped output, $[a, b]$ is the clipping interval and $\beta$ a positive hyperparameter controlling the steepness of softplus function. In these works a pre-specified $[a, b]$ range was used and

---

[5]We used this notation to distinguish it between the ELBO of the S-IntroVAE which we still refer to that simply as W.

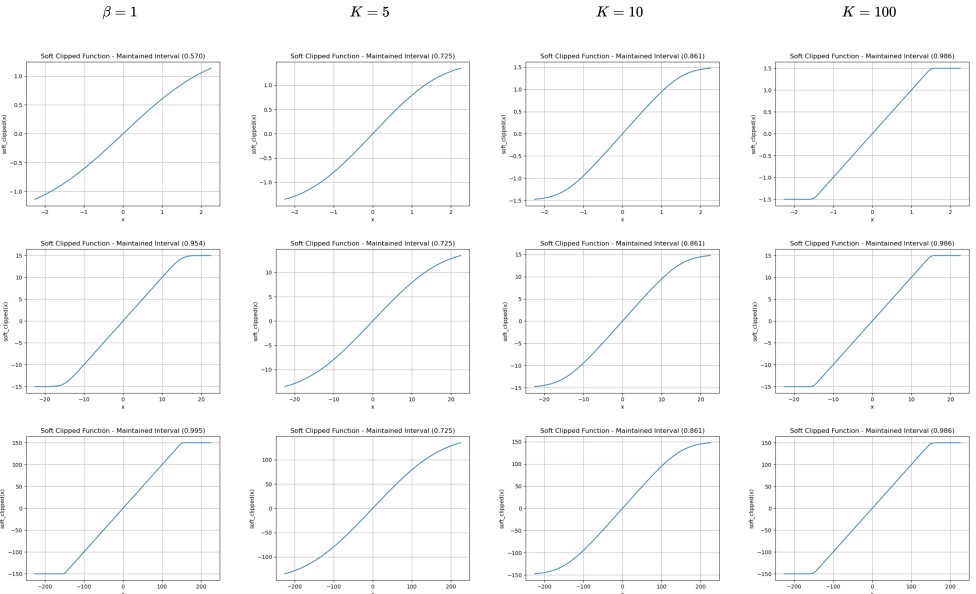

Figure 6: The behavior of soft-clipping depends on the clipping range. Note that when using identical $\beta's$ (e.g. $\beta = 1$) the clipping behavior changes depending on the range (rows). On the other hand, when formulating the $\beta$ as a function of the range and the $K$ hyperparameter the behavior remains consistent. Increasing the $K$ (columns) leads to retaining a bigger portion of the original clipping range.

naturally finding the optimal $\beta$ hyperparameter for softplus is subject to proper fine-tuning. In practice, the default option of $\beta = 1$ was used in both studies.

In our case, different clipping intervals are applied to each latent dimension, that is $[a_j, b_j]$ for each $j^{\text{th}}$ latent dimension. The $[a_j, b_j]$ interval was determined based on the minimum and maximum variance of the prior's modes in each latent dimension as emerged during the VAE pre-training stage. Based on that, identifying the optimal $\beta_j$'s through manual fine-tuning is not a feasible option. Towards overcoming this challenge we model the $\beta_j$'s as:

$$\beta_j = \frac{K}{b_j - a_j}, \tag{49}$$

with $K$ being a controllable hyperparameter. Based on these, we derive the $K$ such that:

$$\frac{f_c(b_j) - f_c(a_j)}{b_j - a_j} \geq \rho, \tag{50}$$

with $\rho \in (0, 1)$. Intuitively the (50) suggests that the initial range should be proportionally maintained post the soft-clipping. The maintained proportion is controlled by $\rho$. Developing (50) based on the soft-clipping function defined in (48) we get:

$$f_c(b_j) - f_c(a_j) \geq \rho \cdot (b_j - a_j)$$

$$b_j - \frac{1}{\beta_j} \cdot \log(1 + \exp(\beta_j \cdot (b_j - b_j))) + \frac{1}{\beta_j} \cdot \log(1 + \exp(\beta_j \cdot (a_j - b_j)))$$
$$-a_i + \frac{1}{\beta_j} \cdot \log(1 + \exp(\beta_j \cdot (a_j - b_j))) - \frac{1}{\beta_j} \cdot \log(1 + \exp(\beta_j \cdot (a_j - a_j))) \geq \rho \cdot (b_j - a_j)$$

$$b_i - \frac{1}{\beta_j} \cdot \log(2) + \frac{1}{\beta_j} \cdot \log(1 + \exp(\beta_i \cdot (a_j - b_j))) - a_j + \frac{1}{\beta_j} \cdot \log(1 + \exp(\beta_j \cdot (a_j - b_j))) - \frac{1}{\beta_j} \cdot \log(2)$$
$$\geq \rho \cdot (b_j - a_j)$$

$$(b_j - a_j) - \frac{2}{\beta_j} \cdot \log(2) + \frac{2}{\beta_j} \cdot \log(1 + \exp(\beta_j \cdot (a_j - b_j)))$$
$$\geq \rho \cdot (b_j - a_j)$$

$$\beta_j \cdot (1 - \rho) \cdot (b_j - a_j) \geq \log(4) - 2\log(1 + \exp(\beta_j \cdot (a_j - b_j))).$$
$$\tag{51}$$

We can derive $K$ using the formulation defined in (49) as:

$$(1 - \rho) \cdot K \geq \log(4) - 2\log(1 + \exp(-K)). \tag{52}$$

Note that the $K$ only depends on $\rho$ and therefore can be tuned for all latent dimensions simultaneously irrespectively of the soft-clipping range $[a_j, b_j]$ (see Fig. 6). In our study, we used a $\rho$ of 0.85 and found that $K = 10$ satisfies the condition (52). In other words, having an adapting $\beta_j = \frac{10}{b_j - a_j}$ guarantees that at least 85% of the initial range is maintained, post-clipping, in all latent dimensions. Alternatively, our $\beta$-adapting formulation can be interpreted as a mechanism where the soft-clipping function maintains the same average rate of change in all latent dimensions, as suggested by (50). Finally, the adaptive clipping function $f_c$ becomes:

$$f_c(\text{logvar}_j) = \text{logvar}_j - \frac{1}{\beta_j} \cdot \log(1 + \exp(\beta_j \cdot (\text{logvar}_j - b_j)))$$
$$+ \frac{1}{\beta_j} \cdot \log(1 + \exp(\beta_j \cdot (a_j - \text{logvar}_j))). \tag{53}$$

### C.2 Losses in S-IntroVAE with trainable prior

Let $x_{\text{real}}$ a real sample and $z_\lambda \sim p_\lambda(z)$. The encoder, the decoder, and the prior players minimize the $L_E$, $L_D$ and $L_P$ losses respectively which write as:

$$L_E(x_{\text{real}}, \lambda) = \beta_{\text{rec}} \cdot L_{\text{rec}}(x_{\text{real}}) + \beta_{\text{KL}} \cdot L_{\text{KL}}(x_{\text{real}}) + \frac{1}{\alpha} \cdot \exp(-\alpha \cdot (\beta_{\text{rec}} \cdot L_{\text{rec}}(D(z_\lambda)) + \beta_{\text{neg}} \cdot L_{\text{KL}}(D(z_\lambda)))),$$

$$L_D(x_{\text{real}}, z_\lambda) = \beta_{\text{rec}} \cdot L_{\text{rec}}(x_{\text{real}}) + \cancel{\beta_{\text{KL}} \cdot L_{\text{KL}}(x_{\text{real}})} + \gamma \cdot (\gamma_\rho \cdot \beta_{\text{rec}} \cdot L_{\text{rec}}(\text{sg}(D(z_\lambda))) + \beta_{\text{KL}} \cdot L_{\text{KL}}(D(z_\lambda))),$$

$$L_P(x_{\text{real}}, z_\lambda) = \cancel{\beta_{\text{rec}} \cdot L_{\text{rec}}(x_{\text{real}})} + \beta_{\text{KL}} \cdot L_{\text{KL}}(x_{\text{real}}) + \gamma \cdot (\cancel{\beta_{\text{rec}} \cdot L_{\text{rec}}(\text{sg}(D(z_\lambda)))} + \beta_{\text{KL}} \cdot L_{\text{KL}}[6](D(z_\lambda))),$$
$$\tag{54}$$

---

[6]When computing this particular KL term we only propagate the gradient for prior as a source while applying the `sg` operator for prior as a target.

where $D(z_\lambda)$ is the fake sample generated from decoding the latent $z_\lambda$, while $L_{\text{rec}}$ and $L_{\text{KL}}$ the reconstruction and the KL losses respectively. Both $L_E$ and $L_D$ are identical to the original S-IntroVAE (Daniel & Tamar, 2021) with $\gamma_\rho$ a hyperparameter also set to $1e^{-8}$. Note that the crossed-out terms do not affect the optimization, as they are constant with respect to the network being updated (e.g., the reconstruction losses are constant with respect to the prior when minimizing the $L_P$).

## C.3 Responsibilities regularization

In this subsection, we provide the theoretical motivation behind the responsibilities regularization which we utilize to discourage the formation of inactive prior modes. The notion of inactivity describes a prior mode that contributes negligibly in supporting the aggregated posterior compared to other more dominant modes. Sampling from inactive prior modes leads to unconstrained generation, which may negatively impact generation performance. To this end, analyzing the minimization behavior of the $L_{\text{KL}}(x_{\text{real}})$ terms in (54) is key to avoiding and/or eliminating the inactive prior modes as these are the terms that induce fitness between the real aggregated posterior and the prior.

Let $q(z|x_s) = \mathcal{N}(z|\mu_s, \sigma_s^2 I)$ be the posterior distribution of the a sample $x_s$, an M-modal prior distribution $p_\lambda(z) = \sum_{i=1}^{M} w_i \cdot \mathcal{N}(z|\mu_i, \sigma_i^2 I)$ and a uni-modal prior distribution $p_i(z) = \mathcal{N}(z|\mu_i, \sigma_i^2 I)$ corresponding to the $i^{\text{th}}$ mode of $p_\lambda(z)$ distribution.

According to the notation defined above, the $L_{\text{KL}}(x_s)$ approximates the KL divergence between the uni-modal posterior $q(z|x_s)$ and the multi-modal prior $p_\lambda(z)$ (i.e., $\text{KL}[q(z|x_s)||p_\lambda(z)]$) as:

$$\text{KL}[q(z|x_s)||p_\lambda(z)] \approx \frac{1}{T} \cdot \sum_{t=1}^{T} \log \frac{q(z_s^t|x_s)}{p_\lambda(z_s^t)} = L_{\text{KL}}(x_s), \tag{55}$$

using $T$ MC samples with $z_s^t \sim \mathcal{N}(z|\mu_s, \sigma_s^2 I)$. Similarly we define the $L_{\text{KL}}^i(x_s)$ as the approximation of the KL divergence between the uni-modal posterior $q(z|x_s)$ and the $i^{\text{th}}$ prior component $p_i(z)$ (i.e., $\text{KL}[q(z|x_s)||p_i(z)]$) as:

$$\text{KL}[q(z|x_s)||p_i(z)] \approx \frac{1}{T} \cdot \sum_{t=1}^{T} \log \frac{q(z_s^t|x_s)}{p_i(z_s^t)} = L_{\text{KL}}^i(x_s), \tag{56}$$

For simplicity, we now assume that $T = 1$ and drop the index $t$ for notational brevity, that is we refer to the $z_s^1$ simply as $z_s$.

### C.3.1 Responsibilities computation - encoder update

First, let us analyze the minimization behavior from the encoder's perspective. For a single MC sample $z_s$, the $L_{\text{KL}}(x_s)$ can be computed as:

$$
\begin{aligned}
L_{\text{KL}}(x_s) &= \log q(z_s|x_s) - \log p_\lambda(z_s) \\
&= \log \mathcal{N}(z_s|\mu_s, \sigma_s^2 I) - \log \sum_{i=1}^{M} w_i \cdot \mathcal{N}(z_s|\mu_i, \sigma_i^2 I).
\end{aligned}
\tag{57}
$$

Based on that, we can now compute the derivative of $L_{\text{KL}}(x_s)$ above with respect to $z_s$ as:

$$\frac{\partial L_{\text{KL}}(x_s)}{\partial z_s} = \frac{1}{\cancel{\mathcal{N}(z_s|\mu_s, \sigma_s^2 I)}} \cdot \cancel{\mathcal{N}(z^t|\mu_s, \sigma_s^2 I)} \cdot \frac{\mu_s - z_s}{\sigma_s^2} - \sum_{i=1}^{M} \frac{w_i \cdot \mathcal{N}(z_s|\mu_i, \sigma_i^2)}{\sum\limits_{l=1}^{M} w_l \cdot \mathcal{N}(z_s|\mu_l, \sigma_l^2)} \cdot \left(\frac{\mu_i - z_s}{\sigma_i^2}\right)$$

$$= \frac{\mu_s - z_s}{\sigma_s^2} - \sum_{i=1}^{M} c_i^s \cdot \left(\frac{\mu_i - z_s}{\sigma_i^2}\right) \tag{58}$$

$$= \sum_{i=1}^{M} \overset{1}{\cancel{c_i^s}} \cdot \frac{\mu_s - z_s}{\sigma_s^2} - \sum_{i=1}^{M} c_i^s \cdot \left(\frac{\mu_i - z_s}{\sigma_i^2}\right),$$

with $c_i^s = \frac{w_i \cdot \mathcal{N}(z_s|\mu_i, s_i^2)}{\sum\limits_{l=1}^{M} w_l \cdot \mathcal{N}(z_s|\mu_l, s_l^2)}$ denoting the responsibility of mode $i$ to $z_s$ of the sample $x_s$.

Similarly we can calculate the derivative of $L_{\text{KL}}^i(x_s)$ with respect to $z_s$ as:

$$\frac{\partial L_{\text{KL}}^i(x_s)}{\partial z_s} = \frac{\mu_s - z_s}{\sigma_s^2} - \frac{\mu_i - z_s}{\sigma_i^2}. \tag{59}$$

Based on Eqs. 58 and 59 we conclude that:

$$\frac{\partial L_{\text{KL}}(x_s)}{\partial z_s} = \sum_{i=1}^{M} c_i^s \cdot \frac{\partial L_{\text{KL}}^i(x_s)}{\partial z_s}. \tag{60}$$

The decomposition provided above reveals the effect that responsibilities of each prior component have when fitting uni-modal posterior into multi-modal prior distributions. More specifically, it is shown that $z_s$ minimizes the $L_{\text{KL}}$ by seeking the prior modes according to the responsibilities $c_i^s$. Motivated by this, we define the expected responsibility of mode $i$ to the real aggregated posterior as:

$$c_i = \mathbb{E}_{x \sim p_{\text{data}}(x)} \mathbb{E}_{z \sim q_\phi(z|x)} \left[ \frac{w_i \cdot \mathcal{N}(z|\mu_i, \sigma_i^2 I)}{\sum_{l=1}^{M} w_l \cdot \mathcal{N}(z|\mu_l, \sigma_l^2 I)} \right]. \tag{61}$$

### C.3.2 Inactive modes and vanishing gradients - prior update

When computing the derivative of the $L_{\text{KL}}(x_s)$ concerning the contribution $w_i$ we will need to take into account that sum of all contributions has to be 1. To ease the computation we can model $w_i = \frac{e_i}{\sum\limits_{l=1}^{M} e_l}$, where $e_i$ is a non-negative real number realizing the unnormalized probability mass of the $i^{\text{th}}$ component, and compute the derivative with respect the normalized energy $e_i$. Based on that :

$$
\begin{aligned}
\frac{\partial L_{\mathrm{KL}}(x_s)}{\partial e_i} &= -\frac{\partial \log \sum_{l=1}^{M} w_l \cdot \mathcal{N}(z_s|\mu_l, \sigma_l^2 I)}{\partial e_i} \\
&= -\frac{1}{\sum_{l=1}^{M} w_l \cdot \mathcal{N}(z_s|\mu_l, \sigma_l^2 I)} \cdot \left( \frac{1}{\sum_{l=1}^{M} e_l} \cdot (1 - w_i) \cdot \mathcal{N}(z_s|\mu_i, \sigma_i^2 I) \right. \\
&\quad \left. - \frac{1}{\sum_{l=1}^{M} e_l} \cdot \sum_{\substack{l=1 \\ l \neq i}}^{M} w_l \cdot \mathcal{N}(z_s|\mu_l, \sigma_l^2 I) \right) \\
&= -\frac{1}{\frac{1}{\cancel{\sum_{l=1}^{M} e_l}} \cdot \sum_{l=1}^{M} e_i \cdot \mathcal{N}(z_s|\mu_l, \sigma_l^2 I)} \cdot \frac{1}{\cancel{\sum_{l=1}^{M} e_l}} \cdot \left( \mathcal{N}(z_s|\mu_i, \sigma_i^2 I) \right. \\
&\quad \left. - \sum_{l=1}^{M} w_l \cdot \mathcal{N}(z_s|\mu_l, \sigma_l^2 I) \right) \\
&= \frac{1}{\sum_{l=1}^{M} e_i \cdot \mathcal{N}(z_s|\mu_l, \sigma_l^2 I)} \cdot \left( \sum_{l=1}^{M} w_l \cdot \mathcal{N}(z_s|\mu_l, \sigma_l^2 I) - \mathcal{N}(z_s|\mu_i, \sigma_i^2 I) \right).
\end{aligned}
\tag{62}
$$

The result above aligns with our intuition as it suggests that given a latent $z_s$ the energy $e_i$ corresponding to the unnormalized contribution of $i^{\text{th}}$ component increases if it is more likely to have been sampled from that mode compared to the MoG prior, and vice versa.

Similarly, we compute the derivatives of $L_{\mathrm{KL}}(x_s)$ with respect with respect to $\mu_i$ and $\sigma_i$ corresponding to the mean and the standard deviation of the $i^{\text{th}}$ prior component respectively. In this case the gradient steps write as:

$$
\begin{aligned}
\frac{\partial L_{\mathrm{KL}}(x_s)}{\partial \mu_i} &= -c_i^s \cdot \frac{z_s - \mu_i}{\sigma_i^2} \text{ and} \\
\frac{\partial L_{\mathrm{KL}}(x_s)}{\partial \sigma_i} &= -c_i^s \cdot \frac{(z_s - \mu_i)^2 - \sigma_i^2}{\sigma_i^3}.
\end{aligned}
\tag{63}
$$

The derivatives above reveal the behavior of the individual prior components in the presence of inactive modes. In particular, an inactive mode $i$ manifests as low $c_i$ responsibility (i.e., $c_i^s$ close to zero for all real sample $x_s$), due to insufficiently supporting the real aggregated posterior relative to other, more dominant modes. Consequently, a vanishing gradient issue arises, where the mean and the standard deviation of the inactive mode $i$ are not updated (towards supporting the posterior) as indicated by 63. On the other hand, the unnormalized contributions of the inactive modes tend to vanish in favor of other more dominant modes as 62 suggests. Based on these observations, it is clear that in the presence of inactive modes, allowing for learnable contributions enables the prior player to eliminate inactive modes. Conversely, not allowing learnable contributions leaves the prior with inactive modes that cannot adapt to the aggregated posterior, due to their low responsibility and consequently vanished gradients rendering the model prone to unconstrained generation.

### C.4  Exploding variance - prior update

In this subsection, we provide the theoretical motivation behind the log-variance clipping that was used to stabilize training. We now assume a posterior $z_s$ such that:

$$|z_s - \mu_i| = t \cdot \sigma_i \quad \text{with} \quad t \gg 1. \tag{64}$$

The formulation above suggests that the sample $z_s$ is highly unlikely under the Gaussian distribution defined by the $i^{\text{th}}$ prior component, with the parameter $t$ controlling the degree of unlikeliness. Under the assumption of (64) the magnitude of the update rules derived in (63) write as:

$$\begin{aligned}
\left|\frac{\partial L_{\text{KL}}(x_s)}{\partial \mu_i}\right| &= c_i^s \cdot \frac{t}{\sigma_i} \text{ and} \\
\left|\frac{\partial L_{\text{KL}}(x_s)}{\partial \sigma_i}\right| &= c_i^s \cdot \frac{t^2 \cdot \sigma_i^2 - \sigma_i^2}{\sigma_i^3} = c_i^s \cdot \frac{t^2 - 1}{\sigma_i} \approx c_i^s \cdot \frac{t^2}{\sigma_i}.
\end{aligned} \tag{65}$$

The derivation above suggests that the gradient magnitudes of the $\mu_i$ and $\sigma_i$ parameters scale linearly and quadratically with $t$, respectively. Based on this, it is evident that the standard deviation, and therefore the variance, of the prior is more sensitive to explosions in the presence of highly unlikely posterior samples $z_s$, compared to the mean. Note that since the contributions $c_i^s$ are computed relative to all prior modes, it is possible for $z_s$ to be unlikely under the $i^{\text{th}}$ prior component while $c_i^s$ remains non-negligible. Finally given that (64) holds, $\text{sign}(\frac{\partial L_{\text{KL}}(x_s)}{\partial \sigma_i}) = -1$ suggesting that the explosion of variance in the presence of unlikely posterior samples results in increasing the variance towards minimizing the KL divergence.

# D  Additional Details and Results

## D.1  Baseline reproduction for 2D datasets

Due to the inherent randomness involved in evaluating the generation quality, the grid-search-based hyperparameter tuning and the computation of KL-divergence in a Monte-Carlo fashion, in table 4 we compare the baseline performance across five key settings. Namely, the baseline as (i) reported in (Daniel & Tamar, 2021) (ii) reproduced by us using the official code-base (Daniel & Tamar, 2021), reproduced by our code-base computing KL both (iii) in closed-form (c) and (iv) in Monte-Carlo (s) manner and finally (v) replicated by our full pipeline of hyperparameter tuning which can result in selecting different optimal hyperparameter for each dataset (compared to those provided by Daniel & Tamar (2021)). Although computing the KL divergence in a Monte-Carlo fashion is unnecessary for the uni-modal prior (baseline) it is important to verify that both closed and Monte-Carlo-based KL computation lead to comparable performance.

| 2D - Dataset | $\beta_{rec}$ | $\beta_{kl}$ | $\beta_{neg}$ |
|---|---|---|---|
| 8Gaussian | 0.2 (0.2) | 0.3 (0.3) | 0.9 (0.9) |
| 2Spirals | 0.2 (0.2) | 0.05 (0.5) | 0.2 (1) |
| Checkerboard | 0.05 (0.2) | 0.2 (0.1) | 0.8 (0.2) |
| Rings | 0.2 (0.2) | 0.2 (0.2) | 0.6 (1) |

Table 3: Optimal hyperparameter under the standard Gaussian prior for each dataset as found using grid-search and as reported by Daniel & Tamar (2021) (in parenthesis).

| | | S-IntroVAE (SG) | | | | |
|---|---|---|---|---|---|---|
| Source → | | reported | official (reproduced) | ours (reproduced) | | ours (replicated) |
| KL Calculation Mode → | | c | c | c | s | s |
| 8Gaussian | gnELBO ↓ | 1.25 ±0.35 | 0.62 ±0.13 | 0.52 ±0.09 | **0.51** ±**0.15** | **0.51** ±**0.15** |
| | KL ↓ | 1.25 ±0.11 | 1.33 ±0.52 | 1.36 ±0.41 | **1.23** ±**0.11** | **1.23** ±**0.11** |
| | JSD ↓ | **0.96** ±**0.15** | 1.16 ±0.15 | 1.16 ±0.11 | 1.01 ±0.18 | 1.01 ±0.18 |
| 2Spirals | gnELBO ↓ | **5.21** ±**0.04** | 5.47 ± 0.05 | 5.47 ± 0.06 | 5.47 ±0.14 | 6.41 ±0.61 |
| | KL ↓ | **8.13** ±**0.3** | 10.21 ±0.39 | 10.66 ±0.19 | 10.26 ±0.39 | 9.5 ±1.23 |
| | JSD ↓ | **3.37** ±**0.04** | 4.03 ±0.1 | 4.11 ±0.16 | 4.08 ±0.06 | 4.21 ±0.5 |
| Checkerboard | gnELBO ↓ | **4.47** ±**0.29** | 6.28 ±0.56 | 6.22 ±0.80 | 6.33 ±0.75 | 7.21 ±0.12 |
| | KL ↓ | 20.27 ±0.21 | 19.72 ±0.23 | 19.99 ±0.28 | 19.94 ±0.37 | **19.62** ±**0.57** |
| | JSD ↓ | 9.06 ±0.15 | 9.04 ±0.19 | 9.34 ±0.19 | 9.19 ±0.17 | **8.87** ±**0.15** |
| Rings | gnELBO ↓ | 6.3 ±0.08 | 5.81 ±0.06 | **5.8** ±**0.05** | 5.85 ±0.13 | 6.03 ±0.12 |
| | KL ↓ | **9.18** ±**0.33** | 10.67 ±0.5 | 10.75 ±0.29 | 10.89 ±0.45 | 9.99 ±0.59 |
| | JSD ↓ | 4.13 ±0.09 | 4.37 ±0.12 | 4.35 ±0.12 | 4.2 ±0.11 | **4.05** ±**0.15** |

Table 4: Baseline performance across five key settings. For each setting, we report the performance (mean ± standard deviation) over five runs. When reporting the performance for columns 2 to 4 we used the optimal hyperparameters as provided (reproduced) by Daniel & Tamar (2021) (also found in parenthesis in Table 3) whereas, for the 5[th] column, we used the optimal hyperparameters found by our grid-search implementation (replicated). Note that for the 8Gaussian dataset, we found the same optimal hyperparameters leading to identical performance between the 4[th] and the 5[th] columns. The 'c' and 's' refer to closed-form and sample-based computation of KL divergence.

## D.2 MoG ablation on the image experiments

In Table 5 we provide the full ablation on the image generation benchmark suggesting that utilizing a sufficient number of prior modes is crucial for achieving optimal generation and representation learning performance.

| | Model → | S-IntroVAE | S-IntroVAE | | | | | | | |
|---|---|---|---|---|---|---|---|---|---|---|
| | Prior Type → | SG | MoG(10) | | | | MoG(100) | | | |
| | LC Flag → | N/A | ✗ | ✗ | ✓ | ✓ | ✗ | ✗ | ✓ | ✓ |
| | IP Flag → | N/A | ✗ | ✓ | ✗ | ✓ | ✗ | ✓ | ✗ | ✓ |
| **MNIST** | $r_{\text{entropy}}$ | 0 | 10 | 0 | 1 | 100 | 10 | 10 | 1 | 10 |
| | Entr. | 0 | $0.966_{\pm0.008}$ | $0.952_{\pm0.014}$ | $0.948_{\pm0.009}$ | $0.988_{\pm0.001}$ | $0.892_{\pm0.002}$ | $0.882_{\pm0.001}$ | $0.882_{\pm0.002}$ | $0.853_{\pm0.004}$ |
| | FID (GEN) ↓ | $1.414_{\pm0.025}$ | $1.38_{\pm0.049}$ | $1.356_{\pm0.1}$ | $1.427_{\pm0.02}$ | $1.365_{\pm0.031}$ | $1.322_{\pm0.025}$ | $1.352_{\pm0.052}$ | $1.32_{\pm0.061}$ | $\mathbf{1.309}_{\pm\mathbf{0.027}}$ |
| | FID (REC) ↓ | $1.503_{\pm0.031}$ | $1.488_{\pm0.072}$ | $1.51_{\pm0.049}$ | $1.629_{\pm0.171}$ | $1.472_{\pm0.069}$ | $\mathbf{1.342}_{\pm\mathbf{0.05}}$ | $1.473_{\pm0.1}$ | $1.363_{\pm0.075}$ | $1.385_{\pm0.081}$ |
| | Recall (GEN) ↑ | $0.565_{\pm0.003}$ | $\mathbf{0.569}_{\pm\mathbf{0.001}}$ | $0.545_{\pm0.006}$ | $0.554_{\pm0.007}$ | $0.552_{\pm0.002}$ | $0.562_{\pm0.003}$ | $0.553_{\pm0.008}$ | $0.556_{\pm0.003}$ | $0.557_{\pm0.001}$ |
| | Precision (GEN) ↑ | $0.522_{\pm0.004}$ | $0.533_{\pm0.002}$ | $\mathbf{0.563}_{\pm\mathbf{0.012}}$ | $0.54_{\pm0.01}$ | $0.553_{\pm0.005}$ | $0.55_{\pm0.005}$ | $0.556_{\pm0.001}$ | $0.561_{\pm0.005}$ | $0.562_{\pm0.005}$ |
| | 2K-SVM ↑ | $0.93_{\pm0.001}$ | $0.956_{\pm0.002}$ | $\mathbf{0.972}_{\pm\mathbf{0.001}}$ | $0.957_{\pm0.002}$ | $0.959_{\pm0.002}$ | $0.961_{\pm0.001}$ | $0.97_{\pm0.004}$ | $0.962_{\pm0.002}$ | $\mathbf{0.972}_{\pm\mathbf{0.002}}$ |
| | 10K-SVM ↑ | $0.93_{\pm0.001}$ | $0.957_{\pm0.002}$ | $\mathbf{0.972}_{\pm\mathbf{0.001}}$ | $0.957_{\pm0.002}$ | $0.958_{\pm0.002}$ | $0.961_{\pm0.001}$ | $0.97_{\pm0.004}$ | $0.962_{\pm0.002}$ | $\mathbf{0.972}_{\pm\mathbf{0.002}}$ |
| | 5-NN ↑ | $0.763_{\pm0.003}$ | $0.866_{\pm0.01}$ | $0.943_{\pm0.005}$ | $0.876_{\pm0.007}$ | $0.842_{\pm0.014}$ | $0.916_{\pm0.004}$ | $0.947_{\pm0.011}$ | $0.92_{\pm0.001}$ | $\mathbf{0.957}_{\pm\mathbf{0.004}}$ |
| | 100-NN ↑ | $0.87_{\pm0.003}$ | $0.897_{\pm0.01}$ | $0.949_{\pm0.006}$ | $0.907_{\pm0.006}$ | $0.885_{\pm0.009}$ | $0.934_{\pm0.002}$ | $0.953_{\pm0.007}$ | $0.935_{\pm0.001}$ | $\mathbf{0.958}_{\pm\mathbf{0.002}}$ |
| **FMNIST** | $r_{\text{entropy}}$ | 0 | 10 | 10 | 0 | 1 | 0 | 10 | 10 | 10 |
| | Entr. | 0 | $0.978_{\pm0.004}$ | $0.982_{\pm0}$ | $0.951_{\pm0.007}$ | $0.82_{\pm0.024}$ | $0.931_{\pm0.003}$ | $0.931_{\pm0.001}$ | $0.944_{\pm0.001}$ | $0.903_{\pm0.005}$ |
| | FID (GEN) ↓ | $3.326_{\pm0.039}$ | $2.778_{\pm0.09}$ | $3.019_{\pm0.095}$ | $2.836_{\pm0.089}$ | $2.987_{\pm0.072}$ | $2.785_{\pm0.051}$ | $3.025_{\pm0.139}$ | $\mathbf{2.727}_{\pm\mathbf{0.079}}$ | $2.831_{\pm0.1}$ |
| | FID (REC) ↓ | $3.76_{\pm0.097}$ | $3.102_{\pm0.062}$ | $3.406_{\pm0.036}$ | $3.189_{\pm0.092}$ | $3.339_{\pm0.081}$ | $\mathbf{2.994}_{\pm\mathbf{0.05}}$ | $3.129_{\pm0.095}$ | $3.185_{\pm0.101}$ | $3.511_{\pm0.074}$ |
| | Recall (GEN) ↑ | $0.314_{\pm0.012}$ | $0.348_{\pm0.005}$ | $0.327_{\pm0.004}$ | $0.338_{\pm0.014}$ | $0.336_{\pm0.004}$ | $\mathbf{0.35}_{\pm\mathbf{0.003}}$ | $0.336_{\pm0.007}$ | $0.346_{\pm0.004}$ | $0.341_{\pm0.008}$ |
| | Precision (GEN) ↑ | $0.518_{\pm0.009}$ | $0.556_{\pm0.005}$ | $0.551_{\pm0.003}$ | $0.558_{\pm0.007}$ | $0.560_{\pm0.004}$ | $0.553_{\pm0.005}$ | $0.558_{\pm0.004}$ | $\mathbf{0.576}_{\pm\mathbf{0.006}}$ | $0.574_{\pm0.003}$ |
| | 2K-SVM ↑ | $0.681_{\pm0.001}$ | $0.703_{\pm0.011}$ | $0.681_{\pm0.010}$ | $0.715_{\pm0.005}$ | $0.68_{\pm0.012}$ | $\mathbf{0.731}_{\pm\mathbf{0.003}}$ | $0.695_{\pm0.007}$ | $0.712_{\pm0.005}$ | $0.696_{\pm0.003}$ |
| | 10K-SVM ↑ | $0.731_{\pm0.006}$ | $0.771_{\pm0.004}$ | $0.763_{\pm0.006}$ | $0.775_{\pm0.003}$ | $0.765_{\pm0.002}$ | $\mathbf{0.78}_{\pm\mathbf{0.002}}$ | $0.772_{\pm0.003}$ | $0.778_{\pm0.002}$ | $0.773_{\pm0.002}$ |
| | 5-NN ↑ | $0.425_{\pm0.009}$ | $0.594_{\pm0.016}$ | $0.649_{\pm0.012}$ | $0.604_{\pm0.015}$ | $0.618_{\pm0.013}$ | $0.683_{\pm0.006}$ | $0.693_{\pm0.008}$ | $0.678_{\pm0.006}$ | $\mathbf{0.707}_{\pm\mathbf{0.005}}$ |
| | 100-NN ↑ | $0.606_{\pm0.014}$ | $0.682_{\pm0.014}$ | $0.691_{\pm0.008}$ | $0.69_{\pm0.010}$ | $0.659_{\pm0.009}$ | $0.736_{\pm0.003}$ | $0.729_{\pm0.006}$ | $0.731_{\pm0.003}$ | $\mathbf{0.739}_{\pm\mathbf{0.004}}$ |
| **CIFAR-10** | $r_{\text{entropy}}$ | 0 | 10 | 10 | 10 | 0 | 10 | 100 | 100 | 10 |
| | Entr. | 0 | $0.895_{\pm0.005}$ | $0.886_{\pm0.006}$ | $0.914_{\pm0.012}$ | 0 | $0.839_{\pm0.007}$ | $0.94_{\pm0.002}$ | $0.929_{\pm0.003}$ | $0.511_{\pm0.043}$ |
| | FID (GEN) ↓ | $4.424_{\pm0.064}$ | $4.538_{\pm0.1}$ | $4.876_{\pm0.075}$ | $4.547_{\pm0.079}$ | $4.595_{\pm0.046}$ | $4.465_{\pm0.038}$ | $\mathbf{4.385}_{\pm\mathbf{0.140}}$ | $4.417_{\pm0.031}$ | $4.594_{\pm0.235}$ |
| | FID (REC) ↓ | $4.13_{\pm0.068}$ | $4.379_{\pm0.053}$ | $4.686_{\pm0.143}$ | $4.539_{\pm0.092}$ | $4.519_{\pm0.059}$ | $4.205_{\pm0.091}$ | $\mathbf{4.084}_{\pm\mathbf{0.006}}$ | $4.141_{\pm0.039}$ | $4.585_{\pm0.373}$ |
| | Recall (GEN) ↑ | $\mathbf{0.283}_{\pm\mathbf{0.003}}$ | $0.266_{\pm0.007}$ | $0.253_{\pm0.003}$ | $0.268_{\pm0.002}$ | $0.267_{\pm0.001}$ | $0.281_{\pm0.001}$ | $\mathbf{0.283}_{\pm\mathbf{0.003}}$ | $0.282_{\pm0.008}$ | $0.264_{\pm0.012}$ |
| | Precision (GEN) ↑ | $0.685_{\pm0.004}$ | $0.687_{\pm0.008}$ | $0.689_{\pm0.008}$ | $\mathbf{0.69}_{\pm\mathbf{0.003}}$ | $0.68_{\pm0.003}$ | $0.676_{\pm0.002}$ | $0.679_{\pm0.004}$ | $0.677_{\pm0.007}$ | $0.685_{\pm0.006}$ |
| | 2K-SVM ↑ | $0.245_{\pm0.009}$ | $0.241_{\pm0.005}$ | $0.264_{\pm0.005}$ | $0.246_{\pm0.01}$ | $0.224_{\pm0.003}$ | $0.25_{\pm0.002}$ | $\mathbf{0.271}_{\pm\mathbf{0.006}}$ | $0.26_{\pm0.002}$ | $0.256_{\pm0.003}$ |
| | 10K-SVM ↑ | $0.391_{\pm0.005}$ | $0.385_{\pm0.004}$ | $0.379_{\pm0.002}$ | $0.387_{\pm0.002}$ | $0.365_{\pm0.002}$ | $0.396_{\pm0.003}$ | $\mathbf{0.407}_{\pm\mathbf{0.007}}$ | $0.401_{\pm0.002}$ | $0.396_{\pm0.002}$ |
| | 5-NN ↑ | $0.206_{\pm0.001}$ | $0.175_{\pm0.002}$ | $0.238_{\pm0.004}$ | $0.175_{\pm0.003}$ | $0.174_{\pm0.004}$ | $0.189_{\pm0}$ | $\mathbf{0.239}_{\pm\mathbf{0.005}}$ | $0.196_{\pm0.001}$ | $0.219_{\pm0.002}$ |
| | 100-NN ↑ | $0.308_{\pm0.007}$ | $0.192_{\pm0.010}$ | $0.305_{\pm0.001}$ | $0.186_{\pm0.005}$ | $0.219_{\pm0.016}$ | $0.216_{\pm0.008}$ | $\mathbf{0.32}_{\pm\mathbf{0.005}}$ | $0.259_{\pm0.003}$ | $0.273_{\pm0.004}$ |

Table 5: Quantitative performance on the images datasets. The LC flag refers to mixture component contributions being learnable while the IP flag refers to training the prior (i.e., prior–decoder cooperation scheme). Reported values are mean ± standard error over three runs. The $r_{\text{entropy}}$ row corresponds to the regularization used to obtain the optimal FID(GEN) for each training configuration, where the Entr. row refers to the normalized entropy of the responsibilities where the closer to one its value the more uniformly the aggregated posterior is supported by the prior components.

### D.3 Illustrating the effect of regularizing the entropy of the responsibilities

Regularizing the entropy of the responsibilities, as described in C.3, was essential for avoiding the formation of inactive modes in our prior–decoder cooperation scheme. Here we provide empirical evidence for that choice by analyzing the curves of normalized entropy of the responsibilities for different regularization intensities controlled by the $r_{\text{entropy}}$ hyperparameter and the corresponding FID(GEN) measuring the generation quality. Towards identifying the optimal value we experimented with $r_{\text{entropy}} \in [0, 1, 10, 100]$, however, to enhance the readability of Fig. 7, we omitted the curves for $r_{\text{entropy}} = 1$ as they displayed similar behavior to $r_{\text{entropy}} = 0$.

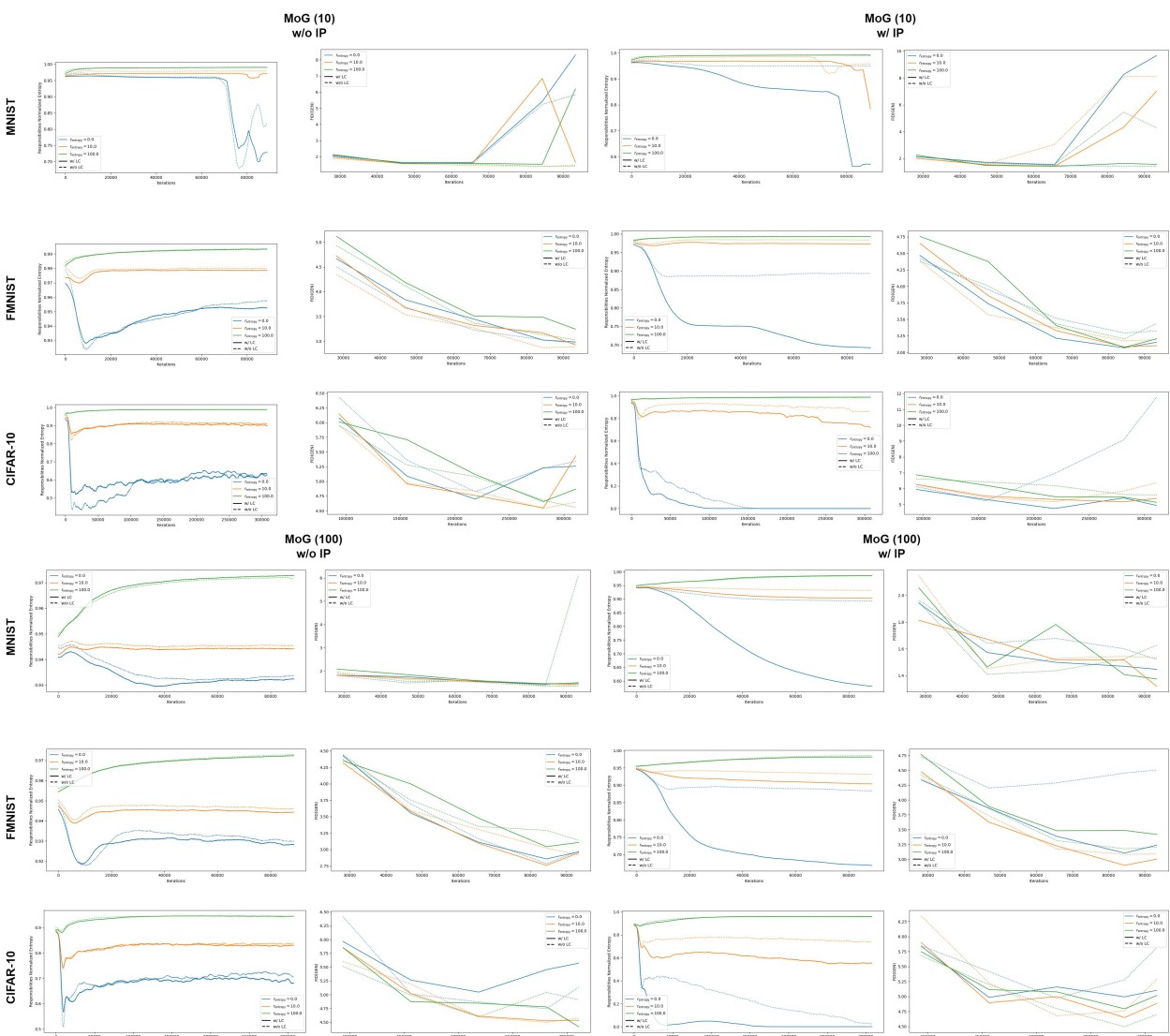

Figure 7: The effect of regularizing the entropy of the responsibilities under the $10-$ and $100-$modal MoG priors.

Inspecting Fig. 7 reveals interesting insights into the effect of responsibilities' regularization. First, it can be seen that different optimal $r_{\text{entropy}}$ are to be expected depending on the prior learning configuration and the datasets as indicated by the FID(curves). Additionally, we observe that the issue of inactive prior mode formation is more pronounced under the IP formulation. The blue lines, representing unregularized responsibilities, tend to converge to a lower level compared to the fixed MoG prior setting. We attribute this

behavior to the prior modes being updated to support the aggregated posterior, which adapts according to a discriminating objective. Interestingly, we also observe that when allowing for learnable contribution (i.e. LC) under the IP generally decreases the entropy of the responsibilities. This observation can be explained by the derivative of KL with respect to the energy contributions as given by 62. More specifically it was shown that the contributions of inactive prior modes tend to decrease in favor of more dominant ones, further reducing the normalized entropy of the responsibilities. Finally, the FID(GEN) curves corresponding to CIFAR-10 dataset highlight the detrimental effect of generating samples from inactive modes. More specifically, when the responsibilities' entropy approaches zero (blue curves in CIFAR-10) the FID(GEN) tends to increase when not allowing for learnable contributions (dotted blue curves). In other words, generating samples from modes that do not support the aggregated posterior (i.e., inactive modes) leads to degraded generation quality.

### D.4 Latent Space Inspection

Here, we provide visualizations of the latent space of S-IntroVAE under the different configurations considered for the image generation task. More specifically we are interested in understanding how allowing for a trainable prior affects the latent space learned in S-IntroVAEs. Overall, the quantitative results suggest that learning the prior during the adversarial training leads to significantly different latent space. In particular, we observe that the prior components are spread more evenly when allowing for trainable prior compared to when fixing it (see Fig. 8).

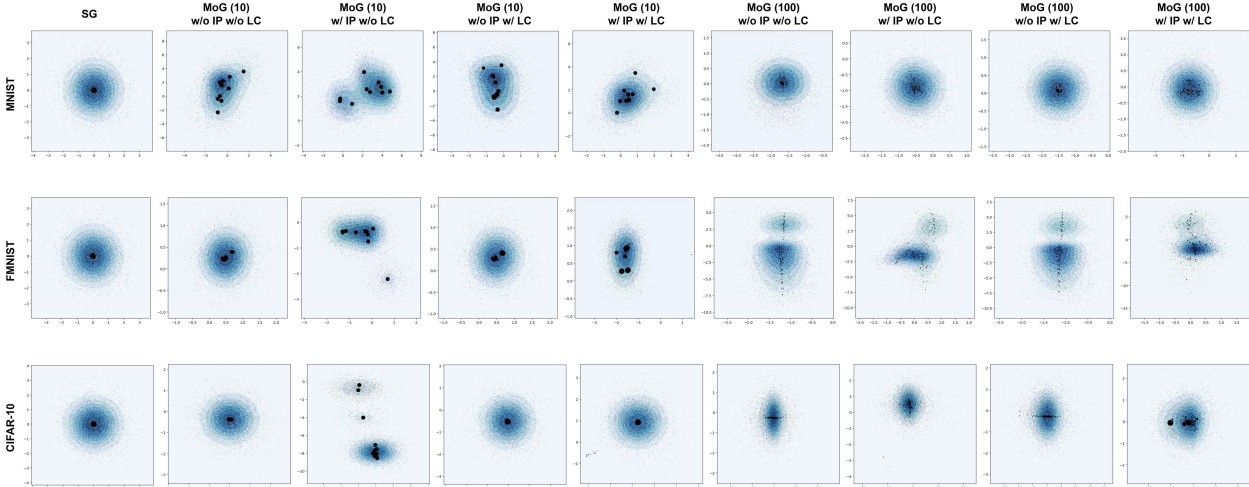

Figure 8: Visualizing the first 2 latent dimensions of S-IntroVAE under different prior configurations along with samples from the aggregated posterior. Different colors correspond to different classes. Note that learning the prior during the adversarial learning leads to significantly different latent space. The black dots refer to the means of the prior components, when applicable (i.e. w/ LC) the size of these dots refers to the contribution of this component in the MoG (e.g. the smaller the size the lower the contribution).

We also employed the t-SNE dimensionality reduction technique to visually inspect how prior learning affects the high-dimensional latent space. The quantitative results indicate that prior learning tends to create better-separated clusters. Although the separation effect is less pronounced when modeling the prior with many components (e.g. 100 vs 10 components), it remains noticeable (see Fig. 9).

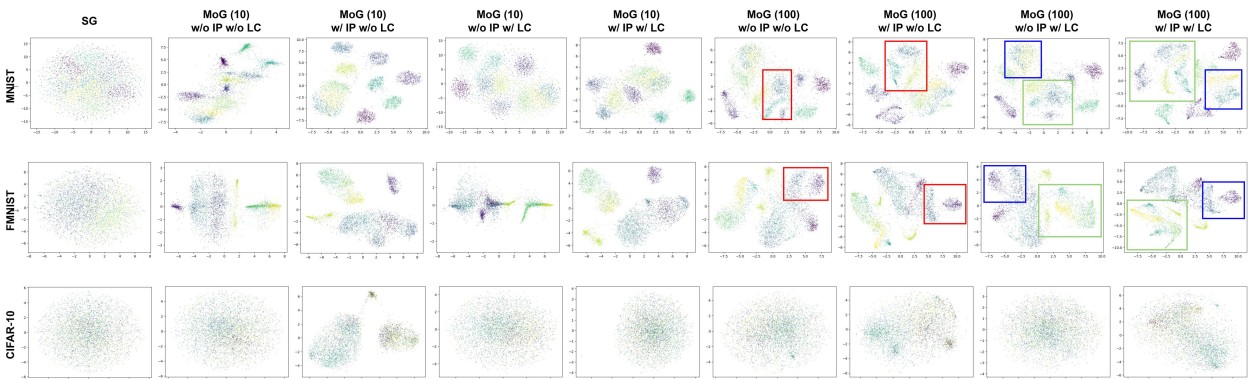

Figure 9: Visualizing the high-dimensional latent space of the aggregated posterior using t-SNE dimensionality reduction technique. Note that learning the prior during the adversarial learning generally leads to better-separated clusters in the latent space. Different colors correspond to different classes.

