# OpenReview forum: "Prior Learning in Introspective VAEs"
_TMLR — Accepted by TMLR_

### Review · Reviewer_kyfR · 2025-03-09

**Summary Of Contributions:**

The work focuses on enabling prior learning for the Soft-Introspective VAE (S-IntroVAE) method. Instead of adopting the standard Gaussian as the VAE prior, the authors employ a finite mixture of Gaussians. However, directly applying the training procedure used in S-IntroVAE can lead to unstable training dynamics and the collapse of the mixture prior, where the Gaussians merge into a single dominant component. To address these issues, the authors introduce adaptive prior variance clipping to stabilize training and mixture weight regularization to prevent prior degeneration.

**Audience:**

No

**Broader Impact Concerns:**

This work focuses on adapting prior distributions for VAE, which is not expected to raise broader social impact concerns.

**Claims And Evidence:**

No

**Requested Changes:**

Please see the weakness part.

**Strengths And Weaknesses:**

### Strengths:
- The scope of this work is well-defined.
- Each methodological component is clearly motivated and properly justified.

### Weaknesses:
- **Presentation of the Proposed Methodology (Section 4) Could Be Improved:**
  - The theoretical results in Section 4.1 do not provide much insight beyond defining the objective function. Unless I am missing key insights, Theorem 2 appears to be a trivial extension of Theorem 1, given that the proof of Theorem 1 is independent of the parametrization of \( p_d \). Additionally, the practical implementation of Proposition 1 deserves to be included in the main text.
  - Many important methodological details are either relegated to the appendix or only explained verbally without proper formulation. For instance, Section 4.2.2 focuses on addressing the prior variance explosion issue. However, it is unclear what specific variance is being referenced. Furthermore, the application of variance clipping (Equation 10) within Algorithm 1 is not explicitly stated.

- **Experimental Details:**
  - It would be helpful if the authors explicitly stated the prior choices of the competitor methods in the experiments.

- **Hyperparameter Tuning:**
  - The proposed training algorithm introduces additional hyperparameters beyond those already required in S-IntroVAE. A systematic approach to tuning these hyperparameters—such as \( K \) for variance clipping, \( r \) for entropy regularization, and the number of mixture components \( M \)—should be elaborated on.

- **Theoretical Claims:**
  - The claim that adaptive variance clipping and responsibility entropy are theoretically motivated seems somewhat overstated. The theoretical justification for these two components is unclear and should be explicitly addressed.

- **Other rather minor Issues**:
  - The mathematical derivations in the appendix lack rigor. In particular, the expectation of certain statistics and their Monte Carlo estimates are often misused. For example, all equalities in Appendix C are mathematically incorrect—KL divergence in this context is not a function of \( z_s \).
  - Several capitalization and formatting errors are present in the References section.

---

> ### Author Response · Authors · 2025-04-25
> **Responding to Reviewer kyfR**
>
> Thank your taking the time to review and provide thoughtful feedback.
>
> **Summary:**
> We found that the overall summary accurately reflects the scope and the main contribution of our work. However, we would like to clarify a critical point.
>
> >To address these issues, the authors introduce adaptive prior variance clipping to stabilize training and mixture weight regularization to prevent prior degeneration.
>
> As discussed in S-4.2.3. the responsibilities regularization encourages a uniform responsibility distribution between the prior modes and the posterior. Note that this is different from the mixture weight regularization. Regularizing only the component contributions (i.e., the mixture weights) does not prevent the emergence of inactive modes. This is because the magnitude of the gradient update for each mode depends on its overall responsibility to the posterior, rather than just its contribution weight (as shown in Eq-62).
>
> &nbsp;
>
> &nbsp;
> **Weaknesses:**
> >The theoretical results in S-4.1 do not provide much insight beyond defining the objective function.
>
> The main insight drawn from Theorem-2. is that the prior--decoder cooperation retains the NE of the vanilla S-IntroVAE. Although the technical proof is a generalization of Theorem-1. that does not undermine the value of Theorem-2. as it gives a theoretically sound way to approach prior learning in S-IntroVAE. Note that in our work we are the first to successfully integrate prior learning in Introspective VAEs as it was previously unknown what would be a reasonable maximization objective for prior learning. To avoid overclaiming on our part we renamed Theorem-2. into  Corollary-1. In relation to that, we also updated Remark-1. to more clearly communicate a minor yet important oversight of Theorem-1.
>
> &nbsp;
> >The practical implementation of Prop. 1 deserves to be included in the main text.
>
> The practical implication of Prop.-1. is now included in the main paper.
>
> &nbsp;
> > Many important methodological details are either relegated to the appendix or only explained verbally without proper formulation. For instance, S-4.2.2 focuses on addressing the prior variance explosion issue. However, it is unclear what specific variance is being referenced.
>
> We revised Sections 4.2.1 and 4.2.2 to enhance clarity and strengthen the logical connection between arguments, supported by more rigorous formulations. We added the C-4. which we point to in S-4.2.2. to formally describe the increasing variance behavior.
>
>
> &nbsp;
> >Furthermore, the application of variance clipping within Alg. 1 is not explicitly stated.
>
> We updated Alg-1. by incorporating feedback from all reviewers.
>
> &nbsp;
> >It would be helpful if the authors explicitly stated the prior choices of the competitor methods in the experiments.
>
> At the beginning of S-5. we included a concrete description for each prior configuration used.
>
> &nbsp;
> > A systematic approach to tuning the additonal hyperparameters should be elaborated on.
>
> The original S-IntroVAE framework consists of 11 tunable parameters that were tuned empirically on a trial-and-error basis. In our case, we introduced 3 additional parameters which we also tuned empirically.
>
> $K$: We argued that $K=10$ is a reasonable choice that post-clipping maintaints more than the 85% percentage of the original range (as discussed in S-C.1.) and found to work well in practice.
>
> $r_\text{entropy}$:  For the image data we experimented with $r_\text{entropy}$ :[0,1,10,100] and identified the optimal configuration with respect to the FID(GEN). The behavior of r is discussed in the last paragraph of S-5.2. while we also added a reference to the full ablation in S-D.3.
> For the 2D dataset, the r was set to 0 as we did not observe the emergence of inactive modes. This different behavior with respect to the r parameter is likely due to the caveat of pairing $\beta_{\text{neg}}$ with the dimensionality of the data as described in the original S-IntroVAE.
>
> $M$: Given the number of modes in each of the 2D datasets, using 64 modes was deemed sufficient. For the experiments on images we experimented with 10 and 100 modes the latter to be superior as discussed in S-5.2.
>
> &nbsp;
> > The claim that adaptive variance clipping and responsibility entropy are theoretically motivated seems somewhat overstated. The theoretical justification for these two components is unclear and should be explicitly addressed.
>
> We updated the S-4.2.1. and -4.2.2 as well as added C-4. to more clearly communicate the connection between the theoretical motivation behind the proposed regularizations. Note that these regularizations are not heuristic but are grounded on the properties of KL minimization and Prop.-1.
>
> &nbsp;
> > The mathematical derivations in the appendix lack rigor.
>
> We updated the derivation and it is now are carried out on the KL loss approximating the KL using a single MC.
>
> &nbsp;
> >Several capitalization and formatting errors are present in the Ref. section.
>
> We corrected the format in Ref.

---

### Review · Reviewer_NfCx · 2025-04-01

**Summary Of Contributions:**

This paper proposes an alternative training scheme for VAEs that learns the prior using an adversarial objective. The motivation is to improve over prior work that learns an VAE using an adversarial objective. Variants of the approach are compared to a standard VAE and a version of prior work that uses an adversarial objective. The experiments are conducted on several low-dimensional density estimation datasets, MNIST, FMNIST, and CIFAR-10. The experiments claim to show that the approach leads to superior density estimation performance and latent separation than relevant alternatives.

**Audience:**

Yes

**Claims And Evidence:**

No

**Requested Changes:**

- First read of the abstract doesn't give a good sense of what the main difference between S-IntroVAE and Introspective VAEs is.
- Introduce the term 'S-IntroVAE' with a reference in the introduction. After reading the abstract and the introduction, it's not clear where the S-IntroVAE is coming from, nor what it signifies. The reference first appears on page 3, despite the term being used on pages 1 and 2.
- L25 in Alg. 1. is too vague, include a pointer to the section or equation that is relevant to implementing it.
- Theorem 2 should be justified with a proof or proof sketch.
- The experiments section should begin with a summary of the goals of the experiments section that connects to the paper's claims.
- The font size on Table 1 is too small, please make it bigger.
- The configuration of the experiments shown in Tables 1 and 2 is not clear. S5.1 introduces three discrete prior settings: SG, fixed MoG, and IP. However, Table 1 does not list them as separate settings. Instead, IP is a row of the table, whereas SG and MoG are columns of the table (and for some reason, SG is in two different columns in Table 1? It looks like maybe one of the columns is supposed to just be the standard VAE, and the rest are S-IntroVAE -- the table needs to make this clearer if so). Altogether this table is very confusing. If the prior has mutually exclusive settings, then they should appear as mutually exclusive values in a cell or some other configuration that shows them being mutually exclusive. I'm having to guess what the experiments actually show here (and the crossed out VAMP in the column title is not helping my understanding, either). Table 2 also appears to be missing the standard VAE results (I cannot tell for sure, the layout is different from Table 1). Fixing all of these issues would be necessary to secure an acceptance recommendation from me.

**Strengths And Weaknesses:**

## Strengths
- Studies a relevant problem
- Clear exposition of prior work
- Experimental settings and scope appear appropriate to potentially support the claims

## Weaknesses
- The biggest weakness is that the construction and presentation of the experiments is too difficult to follow. As a result, I cannot confidently interpret the experimental results, so I cannot confidently say that the experiments support the claims of the paper.
- There are some vagueness in the writing that makes the paper a bit difficult to follow
- The experimental comparison could be more enlightening by comparing to other representation learning approaches besides VAEs (e.g. noise contrastive estimation)

---

> ### Author Response · Authors · 2025-04-25
> **Responding to Reviewer NfCx**
>
> Thank you for taking the time to review our work and provide valuable feedback.
>
> &nbsp;
> **Weaknesses:**
>
> > The biggest weakness is that the construction and presentation of the experiments is too difficult to follow. As a result, I cannot confidently interpret the experimental results, so I cannot confidently say that the experiments support the claims of the paper.
>
> Experimental construction: We have included a paragraph in the beginning of the S-5. summarizing the experimental setup and its motivation. Namely, we define the concrete prior configurations used and clarify that any argument in favor of prior learning requires improvements over both fixed uni-modal (standard Gaussian) and multi-modal (MoG) priors.
>
> Result presentation: We have improved the overall readability of the result by increasing the font size and adding informative labels when necessary. We also revised the result presentation in S-5.1. and -5.2. to better communicate the findings in relation to our claims.
>
> &nbsp;
> > There are some vagueness in the writing that makes the paper a bit difficult to follow
>
> As stated earlier we revised the presentation of the results in  S-5.1. and -5.2. as well as updated S-4.2.1 and -4.2.2. to better connect our arguments to rigorous formulations. We would be happy to address any specific vagueness and clarify to better communicate the message of our paper.
>
> &nbsp;
> > The experimental comparison could be more enlightening by comparing to other representation learning approaches besides VAEs (e.g. noise contrastive estimation)
>
> Our primary objective in this work was to investigate the effect of prior learning in S-IntroVAE. This involved identifying key challenges that hinder effective prior learning, understanding their origins, and addressing them, as detailed in S-4.2. While we agree that expanding the experimental comparison to include other representation learning paradigms (e.g., noise contrastive estimation) is an interesting and natural next step, we believe it lies beyond the scope of the current work. Our focus here is on successfully integrating prior learning into S-IntroVAE and evaluating its impact.
>
> &nbsp;
>
> &nbsp;
> **Requested Changes:**
>
> > First read of the abstract doesn't give a good sense of what the main difference between S-IntroVAE and Introspective VAEs is.
>
> We have revised the abstract to clearly articulate the relationship between S-IntroVAE and Introspective VAEs, as this was previously underspecified.
>
> &nbsp;
> >Introduce the term 'S-IntroVAE' with a reference in the introduction. After reading the abstract and the introduction, it's not clear where the S-IntroVAE is coming from, nor what it signifies. The reference first appears on page 3, despite the term being used on pages 1 and 2.
>
> We have included a reference to S-IntroVAE when it is referred in the text for the first time, namely second to last paragraph of S-1. (p.g. 2).
>
> &nbsp;
> >L25 in Alg. 1. is too vague, include a pointer to the section or equation that is relevant to implementing it.
>
> We have updated the Alg-1. by incorporating feedback from all three reviewers.
>
> &nbsp;
> > Theorem 2 should be justified with a proof or proof sketch.
>
> Taking into consideration feedback from reviewer kyfR we changed Theorem-2 to Corollary-2,  to better reflect its connection toTheorem 1.  A sketch proof of  Corollary-2 is now provided.
>
> &nbsp;
> >The experiments section should begin with a summary of the goals of the experiments section that connects to the paper's claims.
>
> >The font size on Table-1. is too small, please make it bigger.
>
> >The configuration of the experiments shown in Tables-1. and -2. is not clear.
>
>
> Please refer to our response on Weaknesses.
>
> &nbsp;
> > S5.1 introduces three discrete prior settings: SG, fixed MoG, and IP. However, Table-1. does not list them as separate settings. Instead, IP is a row of the table, whereas SG and MoG are columns.
>
> We have reformatted the result tables and updated the caption to better communicate that the IP and LC constitute flags operating on the MoG prior. More specifically, the IP and LC denote trainable prior and learnable component contributions respectively. For example the fourth column of Table-1. corresponds to a trainable MoG (w/ IP) with learnable component contributions (w/ LC), whereas the third column of Table-1. corresponds to a trainable  MoG (w/ IP) with uniform component contributions (w/o LC).
>
> &nbsp;
> > The crossed out VAMP in the column title is not helping my understanding
>
> We dropped the cross-out VAMP formulation while clarifying in the beginning of S-5. and -5.1. that the VAMPPrior is turned into a MoG during the adversarial training.

---

> > ### Comment · Reviewer_NfCx · 2025-04-28
> > **Response to authors**
> >
> > Thank you for the response and updates. The construction and presentation of the experiments is now easier to follow.
> >
> > However, the experimental results do not support the main experimental claim: "the benefit of prior learning in S-IntroVAE in generation and representation learning" (this claim itself is imprecise; I interpreted it to mean that the proposed approach usually achieves better performance). In fact, the results are mixed: the proposed approach does not usually achieve better performance.
> >
> > The experiments on the four 2D datasets show that, actually, prior work usually achieves better performance than any one version of the proposed method. The image generation results show that the proposed approach usually achieves better performance than the baseline, but this effect is somewhat unreliable, as sometimes disabling one of the proposed components (LC) achieves the best performance.
> >
> > In more detail, In Table 1, the baseline (prior work) achieves superior performance in 5/12 of the rows (12 rows = 4 tasks X 3 metrics). The best alternative is with both proposed "flags", achieving superior performance in 3/12 rows. So, while the proposed method can improve performance in some settings, it's less reliable than prior work in these experiments. In contrast, the image generation experimental results are better: the baseline achieves superior performance in 3/24 of the rows, whereas the best alternative (to include both flags), achieves superior performance in 9/24 rows.
> >
> > It's not possible to conclude (at least, without more analysis) that any single version of the proposed method is usually superior to prior work.

---

> ### Author Response · Authors · 2025-04-28
> **Follow up response**
>
> Thank you for taking the time to review our updated version. We are glad that you found the presentation easier to follow.
>
> &nbsp;
>
> We would like to first clarify the distinction between IP and LC. The IP refers to our Intro-Prior formulation in which the prior is learned during the adversarial training. The LC formulation refers to having the component contributions (i.e., the mixture weights) learnable. Having said that, under the IP formulation, the means and variances parameters of the MoG prior are always learnable whereas the component contributions can either also be learnable (w/ LC) or uniform (w/o LC). In our work, we developed the regularizations needed to enable prior learning in S-IntroVAE. Therefore any IP configuration should be regarded as the proposed component, whether w/ or w/o LC.
>
> &nbsp;
> > The experiments on the four 2D datasets show that, actually, prior work usually achieves better performance than any one version of the proposed method.
>
> Tab.1 suggests that an IP-enabled configuration gave optimal forward-generation (KL) performance in 3 out of 4 benchmarks, which as discussed in S-5.1. is intuitive because under the IP both the prior and the decoder push towards minimizing the $\text{KL}[p_\text{data}(x) || p_d^\lambda(x)]$ term. When including also the JSD generation metric an IP-enabled configuration outperforms in 5 out of 8 cases. Although we included gnELBO for completeness, we do not claim any improvements over the inference quality.
>
> In terms of generation performance, if we stick only to the best-performing non-SG configuration, then we effectively need to focus on the w/ IP + w/ LC in which case, although the SG and the IP-enabled configuration both are optimal in 4 out of 8 generation metrics the latter one does so by a larger margin. Based on that an IP-enabled (or the optimal IP-enabled) formulation usually achieves better generation performance which aligns with our claim of improving the generation performance.
>
> Beyond the quantitative measures, the quantitative latent space inspection shown in Fig 4. suggests that the IP configurations better captures the structure of the data which can be also be of interest in certain applications.
>
> &nbsp;
> >The image generation results show that the proposed approach usually achieves better performance than the baseline, but this effect is somewhat unreliable, as sometimes disabling one of the proposed components (LC) achieves the best performance.
>
> We claimed improvements over the generation and the representation quality. Note that the FID incorporates both recall and precision aspects and therefore based on our claims the FIDs and the classification metrics should be the main focus.
>
> Based on that, Tab. 2 suggests that an IP-enabled configuration led to optimal performance in 13 out of the 18 relevant metrics.
> When sticking with a single optimal configuration for each setting we ought to compare the SG (first col.), the w/o + IP w/ LC (fourth col.) and the w/ IP + w/ LC (fifth col.). In this case, each of the SG, fixed MoG and trainable MoG are optimal in 3, 7 and 8 out of 18 metrics respectively.
>
> Note that since non-zero $r_\text{entropy}$ was required to realize the optimal fixed MoG setting (which was part of our contribution - see S-4.2.3) a comparison between the SG and the optimal MoG is of relevance too. In that case, we ought to compare the SG and the w/ IP + w/ LC (fifth col.) where the latter outperforms the former in 4 out of 6 generation metrics and 9 out of 12 metrics (i.e., usually better than the prior work).
>
> When inspecting the t-SNE projection, the IP formulation appears to give rise to more defined clusters compared to the other settings.
>
> &nbsp;
> We acknowledge that the results can appear to be mixed but a closer inspection reveals that learning the prior gives some modest yet existing quantitative improvements over the baselines.
>
> At this point, we would like to kindly ask you to to consider in your evaluation the fact that our work constitutes the first successful attempt of prior learning in Introspective VAEs, which as detailed in the manuscript entailed understanding (S-4.1.1. S-4.1.2)   carefully accounting (S-4.2) for the nature of S-IntroVAE and KL divergence minimization via the proposed regularizations.
>
> Again, we would like to thank you for your response and we are happy to discuss any further questions or clarifications you may have.

---

### Review · Reviewer_5Qnh · 2025-04-14

**Summary Of Contributions:**

This paper proposes prior learning for introspective VAE, the idea of which is to assign low likelihood to unrealistic datapoints. The authors add a modification to the general algorithm of introspective VAEs by

1. Adding a Mixture of Gaussians as a form of prior
2. Modification of the original S-IntroVAE adversarial training objective

**Audience:**

Yes

**Broader Impact Concerns:**

There are not many concerns in relation to this paper, except it contributes to the generation of fake faces etc.

**Claims And Evidence:**

No

**Requested Changes:**

1. Please divide the standard deviations by the $\sqrt{N_{runs}}$ and report the standard error. Reporting standard deviation is not relevant here.
2. I think that something should be done to improve the readability of the results. It is very hard to grasp what the authors are really trying to say from the tables. I believe that among the minimum here is to add a caption telling how to read the results (e.g. row-wise or column-wise).
3. The of Algorithm 1 needs work. Currently it reads as if one defines the procedures in the code, but not one seems to be calling those. Please, revise to make it clear what is called when (I am talking about lines e.g. 7-15. You define a function, but never call).
4. The paper is really missing more convincing benchmarks. The original SoftIntroVAE paper looks at generating high-fidelity images, such as faces. This particular paper looks at CIFAR / MNIST, and Toy data. I request the authors do a proper experimentation and back-to-back comparison between the proposed method and the original SoftIntroVAE.
5. What made me curious, like all works on learning the prior is the capability of the resulting approach to overfit and memorize training data. I suggest the authors to also run the experiment showing whether prior learn does any harm in terms of sample memorization. This is related to my first point "being Bayesian".

**Strengths And Weaknesses:**

Strengths:
* Sound approach
* It is clear what the authors want to do

Weaknesses:
* In a sense, the idea of the prior in Bayesian inference, is to inject prior knowledge and combine it with data. If the prior is learned, I wonder how much is one Bayesian? If the prior is learned, what is the advantage of VAE approach in principle, and and why one wouldn't simply train a regular AE / Masked AE for representation learning. I feel like this part of the paper is rather weak and needs elaboration.
* Another weakness of this paper is toy data. It is a bit hard to get a feeling whether the proposed addition to the introspective VAE is actually helpful.
* Clarity of the result section can be improved
* Statistical comparison is lacking depth

---

> ### Author Response · Authors · 2025-04-25
> **Responding to Reviewer 5Qnh**
>
> Thank you for your time and your thorough feedback.
>
> &nbsp;
> **Weaknesses:**
>
> > The idea of the prior in Bayesian inference, is to inject prior knowledge and combine it with data. If the prior is learned, I wonder how much is one Bayesian?"
>
> Indeed, Classical Bayesian inference requires specifying the prior a priori. In particular, the prior should reflect our prior knowledge about the parameter we are interested in inferring. In contrast, in Empirical Bayesian inference [1] the prior distribution is parametrized and learned from the data.
>
> In the case of VAEs we perform variational inference for latent variables for complex and high-dimensional data such as images. Given the complexity of analyzing or intuiting the latent structure of such data, injecting genuine prior knowledge via the prior becomes challenging. In that sense, the Standard Gaussian (SG) prior distribution commonly used in VAEs is more of a design choice enabling closed-form KL divergence computation and fast sampling rather than true prior knowledge modeling.
>
> In this context, although prior learning in VAEs deviates from being Bayesian in the classical sense, it can be motivated by the absence of meaningful prior knowledge about complex data distributions while remaining Bayesian in the empirical sense.
>
> &nbsp;
> > If the prior is learned, what is the advantage of VAE approach in principle, and why one wouldn't simply train a regular AE / Masked AE for representation learning."
>
> Note that the learnable prior is of finite complexity (i.e. number of training samples is significantly larger than the number of mixture modes). Based on that any objective encouraging fitness to the prior (of finite complexity) acts as a regularization while also providing the means for novel sample generation both of which differentiate prior learning in VAEs from deterministic autoencoding (e.g. regular/Masked AE).
>
> When it comes to representation learning in particular, the regularization imposed by the prior encourages structured latent representations and therefore can reveal the underlying data structures. We added a paragraph in the Introduction section elaborating on this aspect.
>
> &nbsp;
> > Another weakness of this paper is toy data. It is a bit hard to get a feeling whether the proposed addition to the introspective VAE is actually helpful.
>
> We respectfully disagree that experimenting on 2D synthetic data and low-resolution images (e.g. CIFAR10: 3.072 dimensions per sample) constitutes a weakness. We believe that our setting is appropriate to communicate the main messages of our study:
>
> a) Prior learning in Introspective VAEs is possible but requires additional regularization.
> b) Prior learning in Introspective VAEs, when properly regularized, improves the generation performance and the quality of learned representations.
>
> Notably, it is not uncommon for top-tier venues to publish studies on VAEs based solely on low-dimensional synthetic and low-resolution image data. For example, both [2] and [3] are well-received papers published at ICML23 and experiment exclusively on low-dimensional data which suggests that insights drawn even on low-dimensional data are of high relevance to the community. In particular [2] experiments on synthetic 2D and MNIST datasets while [3] uses synthetic 2D, (F)-MNIST, CIFAR10, and mouse cortex cells (1.200 dimensions per sample) datasets.
>
> We also note that in the current work, we approached Introspective VAEs from a representation learning standpoint in which low-dimensional settings are sufficiently revealing. Nonetheless, we consider expanding upon data of higher dimensionality a relevant direction that we leave for future work.
>
> &nbsp;
> > Clarity of the result section can be improved
>
> We have improved the clarity of the results in these aspects:
> a) Added a paragraph at the beginning of the S-5. summarizing the experimental setup and its motivation.
> b) Updated the result presentation in S-5.1. and -5.2.
> c) Improved the overall readability of the result by increasing the font size, adding informative labels, and slight format modifications.
>
>
> &nbsp;
> > Statistical comparison is lacking depth
>
> In our work, we follow standard practices where we report the mean and the standard error across multiple runs.
>
> &nbsp;
>
>
> &nbsp;
> **Requested Changes:**
>
> 1. Applied.
> 2. and 4. Please refer to our response on Weaknesses.
> 3. We have updated the Alg-1. by incorporating feedback from all three reviewers.
> 5. We report the recall and precision metrics in Table-2. and use these to argue that using either learnable or fixed MoG priors does not lead to excessive training sample collapse/memorization when compared to a standard Gaussian prior.
>
> &nbsp;
>
> [1] Carlin, Bradley P., and Thomas A. Louis. "Empirical Bayes: Past, present and future."
>
> [2] Hao, Xiaoran, and Patrick Shafto. "Coupled variational autoencoder."
>
> [3] Kviman, Oskar, et al. "Cooperation in the latent space: The benefits of adding mixture components in variational autoencoders."

---

### Decision · Action_Editor_nHK2 · 2025-05-23

**Recommendation:** Accept with minor revision

**Comment:**

The reviewers were torn on this paper. I have gone over their main points of critique and found that the revision fixes the majority of justified criticisms. A few minor issues would be good to improve upon, so I request a minor revision before accepting the paper.

1. One reviewer points out that it is confusing to have a headline on page 5 followed only by an algorithm. First, I propose to move the algorithm to a later stage in the paper (where it is actually discussed). Secondly, given the length of the algorithm, I would like to have it on its own page without any further content (including headlines).
2. I found remark 1 to raise more questions than it answers. If the implication is important, why is it not discussed in the paper? I would like to see either a bit more detail in the remark, or that the language of the remark is changed to not call the undescribed implication 'important'.
3. The proof sketch in Sec. 4.1 should include a reference to where the full proof can be found.
4. Fig. 2 contains a set of images, which are so small that they cannot be viewed without zooming. Please update the figure to make the images possible to look at.

**Audience:**

The paper extends the S-IntroVAE with a trainable prior, which requires some theoretical and implementation developments over existing work. Given the widespread use of the VAEs and similar models, this paper will most surely have a readership.

**Claims And Evidence:**

The presented method is not always better than state-of-the-art, but all claims are backed up by either empirical or theoretical evidence.